# ENHANCING LANGUAGE MODEL AGENTS USING DIVERSITY OF THOUGHTS

**Vijay Lingam**[†]    **Behrooz Omidvar Tehrani**[†]    **Sujay Sanghavi**[§]    **Gaurav Gupta**[†]
**Sayan Ghosh**[†]    **Linbo Liu**[†]    **Jun Huan**[†]    **Anoop Deoras**[†]

[†]AWS AI Labs    [§]Amazon

`vjlingam@amazon.com`      `omidvart@amazon.com`
`{sujayrs,gauravaz,sghoshnc,linbol,lukehuan,adeoras}@amazon.com`

## ABSTRACT

A popular approach to building agents using Language Models (LMs) involves iteratively prompting the LM, reflecting on its outputs, and updating the input prompts until the desired task is achieved. However, our analysis reveals two key shortcomings in the existing methods: $(i)$ limited exploration of the decision space due to repetitive reflections, which result in redundant inputs, and $(ii)$ an inability to leverage insights from previously solved tasks. To address these issues, we introduce `DoT`[1] (Diversity of Thoughts), a novel framework that a) explicitly reduces redundant reflections to enhance decision-space exploration, and b) incorporates a task-agnostic memory component to enable knowledge retrieval from previously solved tasks—unlike current approaches that operate in isolation for each task. Through extensive experiments on a suite of programming benchmarks (HumanEval, MBPP, and LeetCodeHardGym) using a variety of LMs, `DoT` demonstrates up to a **10%** improvement in Pass@1 while maintaining cost-effectiveness. Furthermore, `DoT` is modular by design. For instance, when the diverse reflection module of `DoT` is integrated with existing methods like Tree of Thoughts (ToT), we observe a significant **13%** improvement on Game of 24 (one of the main benchmarks of ToT), highlighting the broad applicability and impact of our contributions across various reasoning tasks.

## 1  INTRODUCTION

Developing autonomous AI-based frameworks leveraging Large Language Models (LLMs) to solve challenging reasoning and decision-making tasks is an active area of research. Recent research works have focused on utilizing different LLMs as the backbone to develop AI agents that can understand, reason, and reflect on their own actions when solving a task. An AI agent typically performs iterative inference to effectively reason and solve a task by leveraging LLMs and knowledge of tools if available. Examples of such agentic architectures are ReAct (Yao et al., 2023b), Tree of Thoughts (ToT) (Yao et al., 2023a), and Toolformer (Schick et al., 2023), which have explored the in-context learning abilities of LLMs, proposing iterative inference-level algorithms to mimic human-like learning. Such agent architectures promote generalistic approaches to reasoning and learning, which differs from prior reinforcement learning-based solutions introduced by Mnih et al. (2013); Le et al. (2022) that were tailored towards a specific task and demanded extensive compute and data for training.

Building upon ReAct, Reflexion (Shinn et al., 2023) improved the reasoning capabilities in the agent by introducing "self-reflections", a mechanism to steer the model away from previous failed trajectories. Similarly, LATS (Zhou et al., 2023) extended these ideas by enabling multiple reasoning paths and employing search algorithms like Monte Carlo Tree Search (MCTS) to identify optimal solutions. These methods represent significant strides in enhancing LLMs' reasoning capabilities through an iterative feedback.

---

[1]`https://github.com/amazon-science/DiversityOfThoughts`

Table 1: Self-reflection counts and output token usage on a HumanEval subset, highlighting redundancy across problems. "Ref" denotes reflections.

| Problem Name | Total Ref | Unique Ref | #Tokens |
|---|---|---|---|
| 145_order_by_points | 36 | 4 | 5129 |
| 130_tri | 111 | 8 | 14,484 |
| 129_minPath | 84 | 10 | 12,227 |
| 132_is_nested | 36 | 9 | 4292 |
| 84_solve | 96 | 13 | 10,329 |

Table 2: Pass@1 and cost comparison for OpenAI o1 on the LeetCodeHardGym benchmark. `DoT` outperforms Reflexion.

| Method | Pass@1 | Cost |
|---|---|---|
| Base | 45 | $12.75 |
| Reflexion | 62 | $24.24 |
| DoT | **72.5** | $33.47 |

Despite these advancements, current frameworks suffer from few critical limitations: *(i) poor exploration of the decision space, primarily due to repetitive reflections, and (ii) an inadequate memory mechanism.* To depict these shortcomings, we revisited the self-reflections generated by Reflexion and LATS. Specifically, we analyzed the generated self-reflections by these two approaches for the HumanEval dataset (Chen et al., 2021b). Upon careful inspection, we observed that several self-reflections were repetitive. We employed an LLM (GPT-4o) to identify clusters of similar self-reflections and we manually inspected a random sample (see details of prompt used in Appendix A.3). Our analysis revealed that a significant portion of the self-reflections generated by LATS were repetitive and redundant, leading to poor decision-space exploration and excessive token usage. We present key statistics for a representative subset of the HumanEval dataset in Table 1. Similar patterns were observed across other datasets.

Reflexion and LATS incorporate a *local* memory component to store reflections and failed implementations, using it as additional context to improve subsequent generations. However, this memory is reset before tackling the next task. In contrast, Didolkar et al. (2024) demonstrated that recent LLMs can leverage metacognitive abilities to apply prior insights to new tasks, achieving significant performance gains. Their method involves curating in-context examples based on skill exemplars—insights from related tasks—but relies on a predefined set of exemplars, which necessitates additional training data and limits adaptability across diverse tasks.

To address these limitations, we propose Diversity of Thoughts (`DoT`), a novel framework that enhances decision space exploration by (a) reducing redundant reflections and (b) integrating a task-agnostic memory component. `DoT` promotes diverse reasoning attempts using a variety of thought/step sampling strategies, ensuring effective intermediate reflections to guide decision-making. The task-agnostic memory bank dynamically retrieves relevant in-context examples from previously solved tasks, boosting the model's ability to generate informed reasoning paths.

This approach not only reduces redundant computations but also significantly improves efficiency, resulting in a more cost-effective solution. For example, on the LeetCodeHardGym benchmark, `DoT` achieved a 10% gain in Pass@1 (Claude Sonnet 3.5) with a $4\times$ cost reduction compared to LATS (see Section 3.2.1 and Table 6). Additionally, `DoT` demonstrated substantial improvements on OpenAI o1[2] in a preliminary study, outperforming both the base model and Reflexion (see Table 2).

In summary, our main contributions in this paper are mentioned below:

- We identify redundancy as a critical limitation in the existing reflection-based reasoning frameworks (e.g., Reflexion, LATS), as shown anecdotally in Figure 1 and quantitatively in Table 1. Additionally, current frameworks handle each task in isolation, missing the opportunity for cross-task knowledge transfer.

- We propose $(i)$ `DoT`, a novel framework that promotes diversity in reasoning trajectories, and $(ii)$ `DoT-bank`, which further extends `DoT` by incorporating similar trajectories from a task-agnostic memory bank to enhance decision space exploration.

- Our extensive experiments across multiple code generation datasets (HumanEval, MBPP, LeetCodeHardGym) and models demonstrate the effectiveness of `DoT`. We achieve state-of-the-art results, with up to a 10% improvement in Pass@1 on LeetCodeHardGym

---

[2]https://openai.com/index/introducing-openai-o1-preview/

**Reflection-23**

The implementation failed the test case where the list contains both positive and negative numbers. The issue is that the function sorts the numbers based on the sum of their digits and then their actual value. However, it doesn't handle the case where numbers have the same digit sum but different signs. In such cases, the negative number should come before the positive one. To fix this, we need to modify the sorting key to take into account the sign of the number. We can do this by adding a third criterion to the key function that checks if the number is negative. This will ensure that negative numbers come before positive ones when they have the same digit sum.

**Reflection-24**

The implementation failed the test case where the list contains both positive and negative numbers. The issue is with the sorting key function, which calculates the sum of the digits of the absolute value of the number. This causes negative numbers to be treated the same as their positive counterparts. To fix this, we should modify the key function to take into account the sign of the number. We should also consider the order of the numbers in the original list, as the current implementation does not preserve the original order for numbers with the same sum of digits.

**Reflection-27**

The implementation failed the test case where the list contains both positive and negative numbers. The issue is with the sorting key function, which sorts the numbers first by the sum of their digits and then by their value. This approach doesn't correctly handle negative numbers, as it treats them the same as their positive counterparts. To fix this, we need to adjust the key function to properly handle negative numbers. We can do this by adding a third sorting criterion that prioritizes negative numbers over positive ones.

Figure 1: Examples of repetitions in generated self-reflections from LATS on problem `145_order_by_points` in the HumanEval dataset. The words highlighted in red are redundant across Reflection 23, 24 and 27.

for Claude Sonnet 3.5, while reducing token costs by $4\times$ compared to LATS and $1.4\times$ more expensive than Reflexion.

## 2 METHODOLOGY

**Problem Setup.** We briefly review the standard LLM-based reasoning and decision making setup. We are provided with an input $x$ in natural language, along with a pretrained LLM $p_\theta(\cdot)$ parameterized by frozen learned parameters $\theta$. The goal is to generate an output response $y$ that corresponds to a solution for reasoning tasks or a set of actions for decision making. Traditional input-output prompting ($y \sim p_\theta(x)$) leads to sub-optimal performance. Following the observations made by Brown et al. (2020), a flurry of prompting techniques were proposed, which prepend additional input text with specific instructions or few-shot input-output examples to input query $x$. Incorporating such additional context helps to improve reasoning and decision making performance. $\texttt{prompt}_{IO}(x)$ denotes a generic stage in the process of transforming the given input prompt $x$ into output $y$ as: $y \sim p_\theta(\texttt{prompt}_{IO}(x))$. An illustration of this process is shown in Figure 2.

```
System Message:
     You are an AI that only responds with python code, NOT ENGLISH. You
↪    will be given a function signature and its docstring by the user.
↪    Write your full implementation (restate the function signature).

User Message:
def cube_Sum(n: int) -> int:
    """
    Write a python function to find the cube sum of first n even
    ↪   natural numbers.
    """
```

Figure 2: An example of $\texttt{prompt}_{IO}(x)$ on programming tasks. The User Message denotes $x$ in the above example, and the $\texttt{prompt}_{IO}(x)$ prepended a System Message specific to a task.

**Preliminaries.** Our framework's design is inspired by Reflexion (Shinn et al., 2023), which enhances reasoning through iterative interaction with an external API or environment. Reflexion involves three agents—actor ($M_a$), evaluator ($M_e$), and self-reflection ($M_{sr}$)—working cyclically until a termination condition is met. For code generation, the process is as follows:

*(a)* The actor $M_a$ receives input and generates an output (e.g., a code snippet).
*(b)* The evaluator $M_e$ scores this output (e.g., the number of unit tests passed).
*(c)* If the score is low (i.e., code fails), the self-reflection agent $M_{sr}$ diagnoses the issue, suggests a fix, and adds it to the input context along with the failed action. This new input is given to the actor, and the cycle repeats until success or a trial limit is reached.

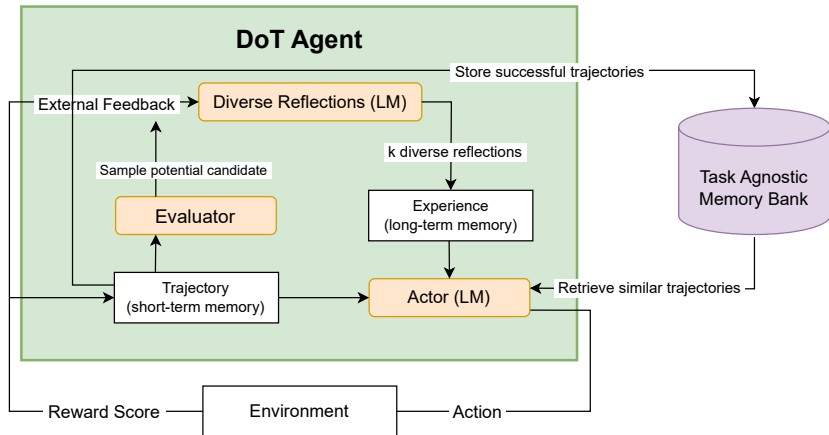

Figure 3: Overview of `DoT` and `DoT-bank`'s architecture. We introduce a new diverse-reflections model to mitigate redundancy in reflections, and a task agnostic memory bank to enable knowledge transfer across tasks.

As noted in Table 1, in several cases, the generated self-reflections are repetitive and redundant, thereby hindering the effective exploration of the decision space.

## 2.1 `DoT`: PROPOSED FRAMEWORK

`DoT` builds upon Reflexion by replacing the self-reflection model with a novel diverse-reflections model, $M_{dr}$. We extend this further with `DoT-bank`, which introduces a task-agnostic memory bank, $MB$. Thus, `DoT` comprises three models: the actor $M_a$ and evaluator $M_e$ (similar to Reflexion) and the diverse-reflections model $M_{dr}$. In `DoT-bank`, $MB$ is additionally leveraged. We provide a detailed description of each component and the workflow that illustrates how these pieces are integrated. An illustration is provided in Figure 3.

**Actor ($M_a$):** The actor is powered by an LLM of choice with specific instructions to generate actions conditioned on the state observations; these actions could be snippets of text, or other actions as per the setting. For example when the task is code generation, the input to the actor is a task description, associated history etc. as a text string, and the output action is a code snippet.

**Evaluator ($M_e$):** Computes a reward score for the action generated by the actor. Depending on the task, $M_e$ can be an external environment (e.g., Python interpreter), a Likert scale rating (Yao et al., 2023a; Zhou et al., 2023), or another LLM (Shinn et al., 2023; Zhou et al., 2023). For example, in the setting of code generation, the reward can be measured by the number of visible or synthetic unit tests passed.

**Diverse-Reflections ($M_{dr}$):** To address redundancy, we propose the *Diverse-Reflections* module, which generates $k$ diverse reflections in one shot using an explicit prompt: $z_i = p_\theta(z_{1...k} \mid \text{DivIO}(x))$. The exact prompt structure ($\text{DivIO}(x)$) is detailed in Figure 5 (see Appendix A.4). Reflections from previous iterations are also included in the context. We investigate alternative approaches for generating $k$ diverse reflections in Section 3.3.3 and find one-shot sampling to be the most effective in terms of both performance and cost. Recent work (Hayati et al., 2024) supports that one-shot sampling can produce semantically diverse outputs. Additionally, we introduce a diversity metric in Section 3.3.1 to quantitatively demonstrate how our $M_{dr}$ module effectively reduces redundancy in generated reflections.

**Task-Agnostic Memory Bank (MB):** Our second contribution is a persistent, task-agnostic memory $MB$ that stores successful trajectories of the `DoT` agent. A trajectory is deemed successful if it passes all visible or synthetic tests. Each entry in $MB$ is indexed by a unique `task-id` and consists of a complete task trajectory. When the actor $M_a$ is invoked, we retrieve $j$ relevant trajectories from $MB$ and include them in its input context. As more tasks are solved, $MB$ grows, facilitating knowledge transfer between tasks. We evaluate our framework in two configurations—`DoT` (without $MB$) and `DoT-bank` (with $MB$)—and analyze the impact of the number of retrieved examples on per-

---

**Algorithm 1** `DoT` and `DoT-bank` Framework

---

**Require:** Dataset $\mathcal{D}$, Retriever, Max Trials $T_{max}$
1: **Initialize:** Actor $M_a$, Evaluator $M_e$, Diverse Reflections Generator $M_{dsr}$,
2: Memory Bank $MB = \emptyset$, Failed Tasks $F = \emptyset$
3:
4: **Phase 1: `DoT`: solving and constructing memory bank**
5: **for** each task $t \in \mathcal{D}$ **do**
6:     Generate $\tau$ using $M_a$ and evaluate using $M_e$
7:     **if** $\tau$ passes evaluation **then**
8:         Add $\tau$ to $MB$
9:     **else**
10:         set i = 0
11:         **while** $t$ not solved or $i < T_{max}$ **do**
12:             Generate $k$ diverse reflections using $M_{dsr}$
13:             Generate $k$ new trajectories using by feeding these reflections and $\tau$ to $M_a$
14:             **for** each generated trajectory $\tau_i$ **do**
15:                 Evaluate $\tau_i$ using $M_e$ and record score
16:                 **if** $\tau_i$ passes evaluation **then**
17:                     Add $\tau_i$ to $MB$, **break the while loop**
18:                 **end if**
19:             **end for**
20:             **if** none of the $k$ trajectories passed **then**
21:                 Select a trajectory at random (weighted by the corresponding score)
22:             **end if**
23:             Increment i
24:         **end while**
25:         **if** no trajectory passes after $T_{max}$ trials **then**
26:             Add $t$ to $F$
27:         **end if**
28:     **end if**
29: **end for**
30:
31: **Phase 2: `DoT-bank`: reattempting failed tasks using memory bank**
32: **for** each failed task $t \in F$ **do**
33:     Retrieve $J$ similar trajectories from $MB$
34:     $M_{a'} \rightarrow$ Inject retrieved trajectories into context of actor
35:     Repeat Phase 1 with $M_{a'}$ and update $MB$ if successful
36: **end for**

---

formance in Section 3.3.3. The orchestration of these agents is presented as `DoT` framework in Algorithm 1. Additional implementation details and a discussion on MB are provided in Appendix A.1.

## 2.2 RELATION TO OTHER METHODS

*Tree-of-Thought (ToT)* (Yao et al., 2023a) extends Chain-of-Thought (CoT) by exploring multiple reasoning paths over intermediate states (also referred to as "thoughts"). It constructs a tree, where each node contains $[x, z_{1...i}]$ and represents a partial solution. Thoughts can either be sampled or proposed using CoT: $z_i \sim p_\theta^{CoT}(x, z_{1...i-1})$. To select the final solution, classic search tree algorithms such as BFS/DFS are employed, combined with a language model-based evaluator that assigns a value to each node.

*LATS* (Zhou et al., 2023) unifies reasoning, acting, and planning under one framework. It replaces the cyclic workflow in Reflexion with a Monte Carlo Tree Search (MCTS) for *proficient exploration* of the decision space. During tree exploration, each node is assigned a value, which is a convex combination of LM based evaluation and self-consistency. A noteworthy detail is that current action is agnostic at a given level of the search tree to previous actions: $a_t^{(i)} \sim p_\theta(s_t)$. We refer the reader to Algorithm 1 in (Zhou et al., 2023) for additional details.

# 3 EXPERIMENTS

In this section, we begin with an overview of the models and datasets used in our experiments. We then describe the experimental setup in detail and conclude with a discussion of the results.

**Summary:** We perform a comprehensive evaluation of our proposed framework, `DoT` and `DoT-bank`, using a wide range of LLMs and benchmarks spanning reasoning and programming tasks. The key findings are: $(i)$ `DoT` and `DoT-bank` consistently outperformed baselines like Reflexion and LATS in terms of accuracy across datasets like HumanEval, LeetCodeHardGym, and MBPP, while being significantly more cost-effective than LATS (see Section 3.2.1). $(ii)$ The performance gains were observed across different LLM architectures and model sizes, with larger improvements seen for less powerful models (see Table 4 and Table 8). $(iii)$ Quantitative analysis (see Section 3.3.1) revealed that `DoT` generates diverse and non-repetitive self-reflections, leading to more effective exploration of the decision space. $(iv)$ Embedding-based retrieval of few-shot examples from a memory bank (`DoT-bank`) further boosted performance. $(v)$ The diversity strategy introduced in `DoT` is also applicable to other reasoning frameworks like Tree of Thoughts, leading to substantial gains on challenging datasets like Game of 24 (see Section 3.3.4).

## 3.1 EXPERIMENT SETUP

**Models:** We conduct a comprehensive evaluation of `DoT` using a diverse suite of both open-source LLMs (Llama-3.1 8B and Llama-3.1 70B) Dubey et al. (2024) and proprietary LLMs (Claude Sonnet 3.5[3], GPT-3.5-Turbo, GPT-4, GPT-4o, GPT-4o-mini[4]). This diverse selection allows for a holistic assessment of our framework's performance across different architectures and model sizes. Our empirical results demonstrate that the effectiveness of our contributions remains consistent, regardless of the specific LLM being used.

**Datasets and Metrics:** In this work, we experiment on a variety of tasks spanning the domains of reasoning and programming. Table 3 presents an overview of the tasks and their corresponding evaluation metrics. For each method, we also report the dollar cost of invoking the respective LLM APIs.

Table 3: Datasets used for Programming and Reasoning tasks.

| Task Type | Dataset Name | Size | Metric |
|---|---|---|---|
| Programming | HumanEval | 164 problems, $\sim$3 visible test cases/problem | Pass@1 |
| Programming | MBPP | 397 sampled problems | Pass@1 |
| Programming | LeetCodeHardGym | 40 *uncontaminated* problems | Pass@1 |
| Reasoning | Game of 24 | 100 puzzles | 0-1 Acc |

To evaluate the programming capabilities of `DoT` and `DoT-bank`, we conduct experiments on HumanEval(Chen et al., 2021a), MBPP (Austin et al., 2021), and LeetCodeHardGym (Shinn et al., 2023) benchmarks, using Pass@1 as our primary metric. During solution generation, only visible or synthetic test cases are used to ensure the validity of Pass@1. The final solution is then assessed on hidden test cases, assigning a Pass@1 score of 1 if all hidden tests are passed, and 0 otherwise.

**Hyperparameters.** We use the hyperparameters recommended by the authors of the respective baselines. For Reflexion, we set max-iterations to $k = 3$ for HumanEval and MBPP, and $k = 5$ for LeetCodeHardGym. For LATS, $k = 8$ is used across all datasets. For `DoT` and `DoT-bank`, we set $k = 3$ for HumanEval and MBPP, and $k = 5$ for LeetCodeHardGym. In Appendix A.2.4, we show that naively increasing $k$ (the total number of generated reflections) in Reflexion has minimal impact on performance, highlighting that `DoT` variants are both more effective and cost-efficient. **Note:** unless specified, the number of retrieved trajectories is 1 for all `DoT-bank` experiments.

---

[3]https://www.anthropic.com/news/claude-3-5-sonnet
[4]https://openai.com/index/hello-gpt-4o/

Table 4: HumanEval Dataset Results Using Visible Test Cases. **Bold** indicates the best result and underscore denotes the second best result. $\Delta_{\text{Base}}$ is the relative improvement over the Base model. `DoT` and `DoT-bank` show consistent improvements in performance.

| Method | Sonnet 3.5 | | | Llama-3.1-8B | | | Llama-3.1-70B | | |
|---|---|---|---|---|---|---|---|---|---|
| | **Pass@1** | $\Delta_{\text{Base}}$ | **Cost** | **Pass@1** | $\Delta_{\text{Base}}$ | **Cost** | **Pass@1** | $\Delta_{\text{Base}}$ | **Cost** |
| Base | 86.59 | - | $0.39 | 59.76 | - | $0.01 | 79.27 | - | $0.04 |
| Reflexion | 88.41 | +1.82 | $0.89 | 72.56 | +12.80 | $0.07 | 87.2 | +7.93 | $0.17 |
| LATS | 88.41 | +1.82 | $7.64 | 74.39 | +14.63 | $2.00 | 84.76 | +5.49 | $4.56 |
| DoT | 91.46 | +4.87 | $1.14 | 73.17 | +13.41 | $0.10 | 89.63 | +10.36 | $0.21 |
| DoT-bank | **93.9** | **+7.31** | $1.67 | **78.66** | **+18.90** | $0.17 | **93.29** | **+14.02** | $0.31 |

## 3.2 EXPERIMENTS & RESULTS

### 3.2.1 PROGRAMMING TASKS

**HumanEval Chen et al. (2021a)** We use visible tests included in the problem doc-strings for all iterative baselines (Reflexion, LATS, `DoT`, and `DoT-bank`). Table 4 compares `DoT` variants against the baselines across two key metrics: accuracy and cost. The results indicate that `DoT` achieves a higher pass@1 rate than LATS, Reflexion, and traditional input-output prompting. Moreover, `DoT` is significantly more cost-effective than LATS, demonstrating that it strikes a more optimal balance between cost and accuracy. `DoT-bank` offers additional performance improvements over `DoT` with gains of up to 4% in pass@1 as shown in Table 4. This highlights the significance of retaining insights from cross tasks. The performance improvement with DoT is independent of the choice of LLM. We observe consistent gains in pass@1 regardless of the underlying LLM, as shown in Table 4. Notably, the performance delta is greater for less powerful models like the Llama-3.1 family.

In Table 5, we compare `DoT` against the baselines using synthetically generated tests. At the time the experiments were done, we discovered a bug in the official LATS implementation, where the "num_success" variable was mistakenly incremented even when an incorrect solution was generated. For comparison, we report both the original and bug-free versions of LATS in Table 5. After fixing the bug, there is a steep 10% drop in Pass@1. Overall, our results suggest that `DoT` delivers superior or competitive performance while being significantly more cost-efficient than LATS.

**LeetCodeHardGym Shinn et al. (2023)** Table 6 compares the performance of `DoT` variants against the baselines using Claude Sonnet 3.5 as base LLM. While LATS incurs a notably high token consumption, leading to increased costs, both Reflexion and LATS fail to deliver any performance gains on this dataset. In contrast, `DoT` effectively explores more diverse reasoning paths, leading to a broader exploration of the decision space, which is reflected in the substantial 7.5% net performance gains. `DoT-bank` further amplifies this improvement with an additional 2.5% gain in the performance. We present additional results using GPT-4o models in Appendix A.2.2.

Table 5: Results for the HumanEval dataset using GPT-3.5-Turbo model. Results with * are obtained from Zhou et al. (2023). $\Delta_{\text{CoT}}$ shows the relative gains w.r.t CoT method.

| Method | Pass@1 | $\Delta_{\text{CoT}}$ |
|---|---|---|
| CoT* | 46.9 | - |
| ReAct* | 56.9 | +10.00 |
| Reflexion* | 68.1 | +21.20 |
| ToT* | 54.4 | +7.50 |
| LATS* | 83.8 | +36.90 |
| LATS (corrected) | 73.9 | +27.00 |
| DoT | **75** | **+28.10** |

**MBPP Austin et al. (2021)**   Since MBPP's problem prompts lack visible test cases by default, we follow Shinn et al. (2023) and use synthetic unit tests to generate potential solutions for all iterative baselines. Consistent with our earlier findings, `DoT` and `DoT-bank` leads to notable performance improvements, with particularly significant gains observed in less powerful LLM backbones as illustrated in Table 8.

To ensure the gains are not random, we repeat select experiments three times and report statistically significant findings in Section 3.3.1. A detailed token usage analysis is provided in Appendix A.2.3.

Table 6: LeetcodeHardGym Dataset Results (Table 1). **Bold** highlights the best result, while underscore marks the second best. $\Delta_{Base}$ indicates the improvement over the Base model.

| Method | Pass@1 | $\Delta_{Base}$ | Cost |
|---|---|---|---|
| Base | 42.5 | - | $0.18 |
| Reflexion | 42.5 | +0.00 | $1.59 |
| LATS | 42.5 | +0.00 | $14.75 |
| DoT | 50.0 | +7.50 | $3.18 |
| DoT-bank | **52.5** | **+10.0** | $3.78 |

Table 7: Accuracy on the Game of 24 Dataset (GPT-4). Augmenting diversity to the existing reasoning frameworks results in significant gains.

| Game of 24 (GPT-4) | Accuracy |
|---|---|
| Base | 7.3% |
| CoT | 4.0% |
| ToT | 69.0% |
| ToT + Diversity | 82.0% (**+13.0%**) |

Table 8: Performance comparison on MBPP using synthetic test cases. DoT and DoT-Bank outperform other methods in Pass@1, while maintaining cost-efficiency across models.

| Method | Sonnet 3.5 | | | Llama-3.1-8B | | | Llama-3.1-70B | | |
|---|---|---|---|---|---|---|---|---|---|
| | Pass@1 | $\Delta_{base}$ | Cost | Pass@1 | $\Delta_{base}$ | Cost | Pass@1 | $\Delta_{base}$ | Cost |
| Base | 77.08 | - | $0.68 | 48.61 | - | $0.02 | 72.80 | - | $0.07 |
| Reflexion | 79.35 | +2.27 | $6.42 | 60.96 | +12.35 | $0.37 | 71.79 | -1.01 | $0.74 |
| LATS | 79.72 | +2.64 | $98.71 | 61.21 | +12.60 | $7.41 | 73.55 | +0.57 | $21.01 |
| DoT | 80.35 | +3.27 | $9.52 | 64.23 | +15.62 | $0.59 | 76.32 | +3.52 | $1.21 |
| DoT-bank | **84.63** | **+7.55** | $9.74 | **66.50** | **+17.89** | $0.71 | **78.34** | **+5.54** | $1.34 |

## 3.3 ADDITIONAL STUDIES

### 3.3.1 QUANTITATIVE ANALYSIS OF GENERATED SELF-REFLECTIONS

To quantify the diversity, we compute the average pairwise cosine similarity across generated self-reflections for each problem within a dataset, reporting both the mean and standard deviation. We use the 'all-Mini-LM-v6' model from SentenceTransformers[5] to generate embeddings for these reflections and tabulate our results in Table 9. A lower similarity score indicates greater diversity.

Table 9: Average pairwise cosine similarity scores for generated reflections across baselines. `DoT` variants produce more diverse and less redundant reflections.

| Dataset + Model | Reflexion | LATS | DoT | DoT-Bank |
|---|---|---|---|---|
| HumanEval + Llama-3.1-70B | $0.83_{\pm 0.09}$ | $0.65_{\pm 0.41}$ | $0.48_{\pm 0.11}$ | $0.48_{\pm 0.12}$ |
| LeetCodeHardGym + Sonnet 3.5 | $0.83_{\pm 0.13}$ | $0.70_{\pm 0.29}$ | $0.62_{\pm 0.13}$ | $0.61_{\pm 0.10}$ |

---

[5]https://sbert.net

### 3.3.2 MEMORY-BANK FEW-SHOT SELECTION'S IMPACT ON PERFORMANCE

Recall from Algorithm 1 that in phase 1, successful trajectories along with the problem's docstring embeddings are stored in the memory bank. In phase 2, for re-attempting failed problems, the docstring embedding is used to retrieve and inject $k$ similar trajectories as in-context examples. To evaluate the impact of retrieved few-shot examples from the memory bank, we conducted an ablative study comparing two strategies: $(i)$ *Random*, which injects $k$ random trajectories from MB into the context of $M_a$, and $(ii)$ *Cosine-sim*, which retrieves the $k$ most similar (closest in embedding space) trajectories based on the cosine similarity of their embeddings. Table 10 demonstrates that embedding-based retrieval consistently outperforms random selection, with performance plateauing at $k = 3$. This highlights that injecting contextually similar examples improves model performance, consistent with recent findings in (Didolkar et al., 2024).

Table 10: Performance on HumanEval with GPT-4o using varying numbers of memory bank examples: Random examples are uniformly sampled, while Cosine-sim examples are retrieved by cosine similarity.

| #ICL Examples | 0 | 1 | 2 | 3 | 4 | 5 |
|---|---|---|---|---|---|---|
| **Random** | 92.68 | 95.12 | **97.56** | 96.95 | 96.34 | 96.95 |
| **Cosine-sim** | 92.68 | 97.56 | 98.17 | **98.78** | 98.17 | 97.56 |

### 3.3.3 GENERATING DIVERSE REFLECTIONS

We analyze two variants for generating diverse reflections and empirically study their performance and the associated costs.

**Iterative Sampling**: To encourage diverse reflections, we condition each generated reflection on all previous ones, incorporating them into the context using a structured prompt. Formally, $z_i \sim P_\theta^{\text{Div}}(z_i \mid x, z_{1...i-1}) \ \forall i \in (1...k)$.

Table 11: Comparison of Sampling Methods for generating diverse reflections on the HumanEval Benchmark. The table shows pass@1 performance and cost for each model using one-shot and iterative sampling methods.

| Sampling Method | GPT-4o-mini | | GPT-4o | | OpenAI o1 | |
|---|---|---|---|---|---|---|
| | **Pass@1** | **Cost** | **Pass@1** | **Cost** | **Pass@1** | **Cost** |
| **One Shot** | **92.07** | **$0.05** | **95.12** | **$1.01** | **99.39** | **$26.75** |
| **Iterative** | 91.00 | $0.07 | 93.29 | $1.57 | 95.73 | $37.37 |

From Table 11, we observe that one-shot sampling for generating diverse reflections consistently outperforms iterative sampling across all models. A recent study by Hayati et al. (2024) further supports our observation. GPT-4o-mini achieves a notable Pass@1 score of 92.07 at a minimal cost ($0.05), making it the most cost-efficient. As model size increases, the cost disparity becomes more pronounced—one-shot sampling is approximately $10 cheaper while delivering a 4% improvement.

Other sampling methods, such as repetitive sampling or using an LLM to filter redundant reflections (similar to our motivation analysis), were not explored due to their higher costs, which compromise cost efficiency.

### 3.3.4 IMPACT OF DIVERSITY ON EXISTING REASONING FRAMEWORKS

The modular design of diversity strategy introduced in this work, aimed at mitigating redundancies in generated steps and thoughts, broadly, is also applicable to the existing reasoning frameworks. As a demonstration, we enhanced the Tree of Thoughts (ToT) framework by incorporating diversity into the thought sampling strategy. This was achieved through explicit prompting and by passing previously generated thoughts through context. We compared this enhanced version with the base

ToT implementation on the challenging Game-of-24 dataset, with the results presented in Table 7. We refer the reader to Appendix A.4.1 for the exact prompts been used. By introducing diversity into the proposed thoughts, we observed a substantial 13% improvement in 0-1 accuracy.

## 4 RELATED WORK

We categorize the literature on LLM reasoning capabilities into three main themes: $(i)$ enhancing step-by-step reasoning abilities, $(ii)$ promoting diversity in the reasoning process, and $(iii)$ incorporating memory mechanisms and external memory to facilitate learning from past experiences.

**Reasoning in LLMs.** Since LLMs were identified as few-shot learners (Brown et al., 2020), several prompting techniques and inference strategies have been proposed to improve their reasoning abilities. Chain of Thought (CoT)(Wei et al., 2022) introduced step-by-step reasoning to enhance problem-solving. Self-Consistency(Wang et al., 2023) extended CoT by leveraging multiple independent CoT runs for improved outcomes. Tree of Thoughts (ToT)(Yao et al., 2023a) employed search algorithms like BFS/DFS with LLM-guided heuristics to further boost performance. ReAct (Yao et al., 2023b) integrates reasoning and action steps, while Reflexion (Shinn et al., 2023) extends this approach by incorporating a "self-reflection" component, building upon the Self-Refine framework (Madaan et al., 2023). More recently, LATS (Zhou et al., 2023) integrated Monte Carlo Tree Search (MCTS) into Reflexion, achieving gains at the expense of higher token usage and costs. However, our analysis revealed that Reflexion and LATS generate redundant self-reflections thereby hindering effective decision space exploration.

**Diversity in Reasoning.** Inducing diversity in LLM reasoning has been explored through methods like DIV-se (Naik et al., 2024), which uses varied prompts and personas (e.g., "Think like Alan Turing" or "Think like a Math Professor"). However, it requires manual persona selection, limiting flexibility. Flow of Reasoning (FoR)(Yu et al., 2024) uses GFlowNet to train LLMs to generate diverse reasoning without predefined personas but involves task-specific fine-tuning. In contrast, our framework, `DoT`, promotes diversity through structured prompts and one-shot sampling, leveraging a task-agnostic memory bank without personas, manual interventions, or task-specific training. This enables `DoT` to explore multiple decision branches and adapt across domains seamlessly.

**LLM Memory.** Memory mechanisms enable LLMs to retain and learn from past experiences, enhancing their ability to reason and adapt. MemoryBank (Zhong et al., 2024) enables LLMs to update and retrieve past interactions to align with user intent. MemoChat (Lu et al., 2023) trains LLMs to efficiently retrieve relevant dialogue history. Reflexion (Shinn et al., 2023) leverages memory in a reinforcement learning manner, with past action described in verbal format. CLIN Majumder et al. (2024) stores causal abstractions in an evolving memory. In contrast, our task agnostic memory bank stores entire agent trajectories retrieving them as in-context examples, enriching decision-making and reasoning.

**External Memory for LLMs.** External memory methods like Buffer of Thoughts (BoT) (Yang et al., 2024) introduce high-level "thought templates" to assist reasoning, while Needle in a Haystack Chaudhury et al. (2024) demonstrates the utility of external memory for tasks requiring long context lengths. Our approach differs by storing complete task trajectories in a dynamic, task-agnostic memory bank, injecting relevant examples to improve reasoning. The memory-bank grows as the agent solves more tasks.

## 5 LIMITATIONS & CONCLUSION

`DoT` and `DoT-bank` significantly reduce costs compared to LATS but remain $1.4\times$ more expensive than Reflexion and up to $8\times$ higher than the base LLM. Iterative reasoning frameworks also suffer from increased latency due to repeated interactions with external environments (e.g., tools, APIs), limiting scalability. While our one-shot sampling mitigates redundancies in generated reflections, it is not entirely foolproof, leaving room for future research on more efficient strategies to generate diverse reflections. This work addresses key limitations in self-reflection-based reasoning frameworks, namely redundant reflections and missed opportunities for cross-task knowledge transfer. Although focused on programming tasks, extending these methods to other reasoning domains presents an exciting avenue for future exploration.

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

# A APPENDIX

## A.1 ADDITIONAL IMPLEMENTATION DETAILS

The memory bank is implemented as a hash map: unique-task-id: complete-task-trajectory. In phase 1 of our DoT algorithm, successful trajectories and the embedding of the task's docstring are stored. In phase 2, when re-attempting failed tasks, we generate an embedding for the task's docstring and retrieve the top $k$ most similar trajectories using cosine similarity. We use Cohere-Embed-V3-English for embedding generation. The $k$ closest trajectories, based on cosine similarity, are considered relevant and serve as in-context examples for the actor agent.

## A.2 ADDITIONAL EXPERIMENTS

### A.2.1 EXPERIMENTS WITH STATISTICAL SIGNIFICANCE

To ensure the reported gains are not random, we repeat a subset of experiments three times and report both average and standard deviation in Table 12. These results further validate DoT's effectiveness.

Table 12: 3-run average Pass@1 on HumanEval and LeetCodeHardGym using Sonnet-3.5. DoT variants show statistically significant improvements.

| Model | HumanEval | LeetCodeHardGym |
|---|---|---|
| Base | $85.57_{\pm 0.94}$ | $40.83_{\pm 2.89}$ |
| Reflexion | $88.21_{\pm 0.35}$ | $41.67_{\pm 1.44}$ |
| LATS | $89.16_{\pm 0.68}$ | $42.50_{\pm 2.50}$ |
| **DoT** | $\mathbf{91.46}_{\pm 0.61}$ | $\mathbf{50.00}_{\pm 2.50}$ |
| **DoT-bank** | $\mathbf{94.10}_{\pm 0.35}$ | $\mathbf{53.33}_{\pm 1.44}$ |

### A.2.2 LEETCODEHARDGYM

We replicate the LeetCodeHardGym experiments using GPT-4o and GPT-4o-mini, and summarize the results in Table 13. DoT achieves comparable or superior performance with a $4\times$ reduction in cost. Notably, DoT-bank outperforms LATS by 12.5% on GPT-4o, while being $2.5\times$ more cost-efficient.

Table 13: **LeetcodeHardGym Dataset Results with Visible Test Cases. Bold** highlights the best result, while underscore marks the second best. $\Delta_{\text{Base}}$ indicates the improvement over the Base model. DoT and DoT-bank consistently enhance performance across various base models.

| Method | GPT-4o-mini | | | GPT-4o | | |
|---|---|---|---|---|---|---|
| | Pass@1 | $\Delta_{\text{Base}}$ | Cost | Pass@1 | $\Delta_{\text{Base}}$ | Cost |
| Base | 15.0 | - | $0.05 | 25.0 | - | $0.18 |
| Reflexion | 17.5 | 2.50 | $0.10 | 27.5 | 2.50 | $1.26 |
| LATS | 20.0 | 5.00 | $0.72 | 32.5 | 7.50 | $13.33 |
| DoT | **22.5** | 7.50 | $0.15 | 32.5 | 7.50 | $3.18 |
| DoT-bank | 22.5 | 7.50 | $0.26 | **45.0** | 20.0 | $5.48 |

### A.2.3 LLM TOKEN USAGE ANALYSIS

Table 14 presents a breakdown of average input and output token usage across different methods. DoT's token usage closely mirrors that of Reflexion, yet it achieves significantly better performance. In contrast, LATS is an order of magnitude more token- and cost-intensive than the other baselines.

Table 14: Token usage and cost analysis for different methods on the HumanEval dataset using Sonnet-3.5.

| Method | Avg Input Tokens | Avg Output Tokens | Avg Cost per task ($) |
|---|---|---|---|
| Base | 223 | 114 | 0.0024 |
| Reflexion | 702 | 221 | 0.0054 |
| LATS | 9370 | 1230 | 0.0466 |
| DoT | 904 | 282 | 0.0070 |
| DoT-bank | 1418 | 395 | 0.0102 |

### A.2.4 IMPACT OF INCREASED ITERATIONS ON REFLEXION PERFORMANCE

Table 15 analyzes the impact of increasing the maximum iterations ($k$) on performance (Pass@1) for the Reflexion method. The table shows that naively increasing the number of reflections has minimal impact on performance, while DoT variants outperform Reflexion at $k = 3$.

Table 15: Performance and Cost Analysis for Different K Values

| Method | K=3 | K=6 | K=8 | K=10 |
|---|---|---|---|---|
| Reflexion | 88.41 ($0.89) | 89.02 ($1.36) | 89.63 ($1.64) | 89.63 ($1.87) |
| DoT | 91.46 ($1.14) | – | – | – |
| DoT-bank | 93.90 ($1.67) | – | – | – |

### A.3 PROMPT TO CLUSTER REFLECTIONS USING GPT-4O

We use the prompt illustrated in Figure 4 to cluster reflections by using GPT-4o. In Table 16 we show an example of how the output from GPT-4o looks like corresponding to the aforementioned prompt.

```
You are provided with a list of sentences, each identified by a unique
↪   ID. Your task is to group these sentences into clusters, where each
↪   cluster contains sentences that convey the same meaning. The
↪   sentences might be phrased differently, but they should express the
↪   same core idea or intent.

Instructions:
Review each sentence carefully, considering synonyms, paraphrasing, and
↪   variations in expression.
Group the sentences that share the same meaning into a single cluster.
For each cluster, list the unique IDs of the sentences included.

Input Format:
A list of sentences with unique IDs.

Output Format:
Cluster IDs along with a concise summary of each cluster, presented in
↪   table format
```

Figure 4: Prompt to cluster reflections

| Cluster ID | Sentence IDs | Summary |
|---|---|---|
| 1 | 0, 1, 2, 3, 4, 5, 6, 7, 8, 9, 10, 11, 12, 13, 14, 15 | The implementation fails to raise a TypeError for non-integer inputs, particularly strings. Type checking needs to be improved to explicitly raise TypeErrors for non-integer inputs. Various methods for fixing include adjusting conditions or using try-except blocks. |
| 2 | 16, 17, 18, 19, 20, 21, 22, 23, 24, 25, 26, 27, 28, 29, 30, 31, 32, 33, 34, 35, 36, 37, 38, 39, 40, 41, 42, 43, 44, 45, 46, 47, 48, 49, 50, 51 | The implementation fails to raise a TypeError for non-integer inputs, and relies on try-except blocks which do not handle all cases. The solution involves adding explicit type checks before calculations to ensure correct error handling. |

Table 16: Clusters of sentences with similar meaning and their summaries

## A.4 PROMPTS USED IN DOT

We use the prompt in Figure 5 to generate $K$ diverse reflection. We use the prompt in Figure 6 to generate function implementations by using the reflections and few-shot examples for the programming tasks.

```
You are a Python programming assistant. You will be given a function
↪  implementation, unit tests, and previously generated reflections.
Write multiple unique and diverse reflections to fix the problem. Each
↪  reflection should follow this structure:

Problem: {a terse description of the identified problem}.
Fix: {proposed fix or hint to fix the identified problem}.

Ensure your reflections are accurate and leverage previous reflections
↪  to avoid repetition. Aim for diversity in your explanations while
↪  prioritizing the correctness of your hints. Add "\n\n" at the end
↪  of each proposed reflection. Only provide the few sentence
↪  descriptions in your answer, not the implementation.
RESTRICT TO {K} UNIQUE REFLECTIONS.
```

Figure 5: Prompt for generating $K$ diverse reflections

## A.4.1 ADDING DIVERSITY TO TREE-OF-THOUGHTS

We use the following prompt shown in Figure 7 to add diversity over Tree-of-thoughts for Game of 24 task (corresponds to results shown in Table 7).

```
Example 1:
[function impl]:
```python
def longest_subarray_with_sum_limit(nums: List[int], target: int) ->
↪   List[int]:
    n = len(nums)
    left, right = 0, 0
    max_length = 0
    current_sum = 0
    result = []
    while right < n:
        current_sum += nums[right]
        while current_sum > target:
            current_sum -= nums[left]
            left += 1
        if right - left + 1 >= max_length:
            max_length = right - left + 1
            result = nums[left:right+1]
        right += 1
    return result
```

[unit test results]:
Tests passing:
assert longest_subarray_with_sum_limit([1, 2, 3, 4, 5], 8) == [1, 2, 3]
assert longest_subarray_with_sum_limit([1, 2, 3, 4, 5], 15) == [1, 2,
↪   3, 4, 5]
assert longest_subarray_with_sum_limit([1, -1, 2, -2, 3, -3], 2) == [1,
↪   -1, 2, -2, 3]
assert longest_subarray_with_sum_limit([], 10) == []
assert longest_subarray_with_sum_limit([], 0) == []
assert longest_subarray_with_sum_limit([], -5) == []
Tests failing:
assert longest_subarray_with_sum_limit([5, 6, 7, 8, 9], 4) == [] #
↪   output: [5]

[self-reflection]:
Problem: The implementation failed the test where no subarray fulfills
↪   the condition. The issue is due to the use of `>=` instead of `>`
↪   in the condition to update the result.
Fix: Change the condition to only update the result when the current
↪   subarray length is strictly greater than the previous longest
↪   subarray length by replacing `>=` with `>` in the condition.

Problem: The current implementation does not handle the case where the
↪   sum of any subarray exceeds the target, leading to incorrect
↪   results.
Fix: Add a condition to check if the sum of the subarray exceeds the
↪   target before updating the result.

Problem: The algorithm might not correctly identify the longest
↪   subarray when multiple subarrays have the same length but different
↪   sums.
Fix: Introduce an additional check to ensure that the subarray with the
↪   sum closest to the target is selected when lengths are equal.

Problem: The code does not consider the possibility of an empty array
↪   or a target that is too small to be achieved by any subarray.
Fix: Implement a preliminary check to return an empty array if the
↪   target is smaller than the smallest element in `nums` or if `nums`
↪   is empty.

___
```

```python
Example 2:
[function impl]:
```python
def longest_subarray_with_sum_limit(nums: List[int], target: int) ->
↪  List[int]:
    n = len(nums)
    left, right = 0, 0
    max_length = 0
    current_sum = 0
    result = []
    while current_sum + nums[right] <= target:
        current_sum += nums[right]
        right += 1
    while right < n:
        current_sum += nums[right]
        while current_sum > target:
            current_sum -= nums[left]
            left += 1
        if right - left + 1 > max_length:
            max_length = right - left + 1
            result = nums[left:right+1]
        right += 1
    return result
```

[unit test results]:
Tests passing:
assert longest_subarray_with_sum_limit([], 10) == []
assert longest_subarray_with_sum_limit([], 0) == []
assert longest_subarray_with_sum_limit([], -5) == []
Tests failing:
assert longest_subarray_with_sum_limit([1, 2, 3, 4, 5], 8) == [1, 2, 3]
↪  # output: list index out of range
assert longest_subarray_with_sum_limit([1, 2, 3, 4, 5], 15) == [1, 2,
↪  3, 4, 5] # output: list index out of range
assert longest_subarray_with_sum_limit([5, 6, 7, 8, 9], 4) == [] #
↪  output: list index out of range
assert longest_subarray_with_sum_limit([1, -1, 2, -2, 3, -3], 2) == [1,
↪  -1, 2, -2, 3] # output: list index out of range

[self-reflection]:
Problem: The implementation fails due to an IndexError caused by
↪  accessing `nums[right]` without bounds checking.
Fix: Add a bounds check for the `right` variable in the `while
↪  current_sum + nums[right] <= target:` loop by modifying the
↪  condition to `while right < len(nums) and current_sum + nums[right]
↪  <= target:`.

Problem: The initial while loop does not account for the scenario where
↪  the right index exceeds the length of the array.
Fix: Introduce a boundary condition to ensure `right` remains within
↪  the array limits during the initial accumulation phase.

Problem: The implementation incorrectly attempts to extend the subarray
↪  beyond the array's bounds, leading to a runtime error.
Fix: Implement a check within the loop to ensure `right` does not
↪  exceed the length of `nums` before accessing `nums[right]`.

Problem: The code does not correctly handle cases where the target sum
↪  is less than any single element in the array, resulting in a
↪  failure to return an empty array.
Fix: Add a preliminary check to return an empty array if no subarray
↪  can possibly meet the target.

END OF EXAMPLES
```

Figure 6: Prompt for generating function implementations by using the reflections and few shot examples.

```
Task: Game of 24 is a mathematical reasoning challenge, where the goal
↪  is to use 4 numbers and basic arithmetic operations $(+ - * /)$ to
↪  obtain 24.

You have the following numbers: {input}

Generate several possible next steps by applying basic arithmetic
↪  operations (+, -, *, /) to pairs of numbers.
Each step should use a unique combination of numbers and a unique
↪  operation that has not been used in the current set of
↪  possibilities.

Possible operations include:
- Addition: a + b
- Subtraction: a - b or b - a
- Multiplication: a * b
- Division: a / b or b / a (provided the result is an integer)

Instructions:
1. Do not repeat any operations that have already been listed.
2. Avoid using the same pair of numbers for the same operation.
3. Provide the result of the operation and show the new set of numbers
↪  after applying the operation.

Example:

Input: 4, 5, 6, 7

Possible next steps:
4 + 5 = 9 (left: 9 6 7)
4 * 5 = 20 (left: 20 6 7)
4 - 5 = -1 (left: -1 6 7)
5 - 4 = 1 (left: 1 6 7)
4 + 6 = 10 (left: 5 7 10)
4 * 6 = 24 (left: 5 7 24)
4 - 6 = -2 (left: -2 5 7)
4 + 7 = 11 (left: 5 6 11)
4 * 7 = 28 (left: 5 6 28)
4 - 7 = -3 (left: -3 5 6)
5 + 6 = 11 (left: 4 7 11)
5 * 6 = 30 (left: 4 7 30)
5 + 7 = 12 (left: 4 6 12)
5 * 7 = 35 (left: 4 6 35)
5 - 7 = -2 (left: -2 4 6)
6 + 7 = 13 (left: 4 5 13)
6 * 7 = 42 (left: 4 5 42)
6 - 7 = -1 (left: -1 4 5)

Input: {input}

Possible next steps:
```

Figure 7: Prompt for obtaining diverse reasoning chains for the Game of 24 task

