# OpenReview forum: "Enhancing Language Model Agents using Diversity of Thoughts"
_ICLR.cc/2025/Conference — ICLR 2025 Poster_

### Official Review · Reviewer_vf4f · 2024-11-01

**Soundness:** 3
**Presentation:** 3
**Contribution:** 4
**Rating:** 8
**Confidence:** 3

**Summary:**

The paper presents an iterative prompting method for LLMs that aims to improve responses by repeatedly reflecting on its previous outputs similar to the recent Reflexion method. The key advance here over the Reflexion method is to encourage diversity in the outputs. The method also incorporates a task-agnostic memory component to enable knowledge retrieval from previously solved tasks.

**Strengths:**

The paper is well laid out and clearly describes the method and justification for it. The method is general and can be widely applied as well as supporting integration with other orthogonal approaches (the example being tree of thoughts). The experiments are varied and well chosen with appropriate choice of base LLMs and tasks. The results show a consistent and clear improvement over other recent SoTA methods: Reflexion, LATS and ToT (where appropriate).

I am particularly impressed with the modular nature of DoT and its generality.

**Weaknesses:**

The weaknesses are, in my opinion, minor. The following things came to mind:
* The description of retrieval of previous trajectories could be a bit clearer, as could the experiment which produces the results in Table 9.
* I am not sure that the t-SNE evaluation in Figure 4 is as robust as it could be. It may show that DoT produces more diverse self-reflections then LATS but as the embedding is done separately for the two sets of reflections it isn't entirely clear that this is the case. I would suggest embedding the two methods together and showing that the DoT reflections are more dispersed within the same space (maybe that is what was done but it isn't clear to me).
* There are lots of results and space is at a premium but things do get less clear towards the end of these, perhaps partly due to the formatting.

**Questions:**

It isn't clear how an evaluator can be an LLM. Is the LLM asked to produce a numeric score? Or is there space for a textual evaluation?

When you say that MB trajectories are picked based on cosine-similarity, do you mean that the K most similar are chosen? Is there a risk that there isn't much diversity in these retrieved trajectories? Or am I missing something?

---

> ### Author Response · Authors · 2024-11-14
>
> We thank the reviewer for their constructive feedback and are glad that they found our work to be well-motivated and well-justified. We are encouraged by the recognition of our carefully chosen experiments and the clarity and consistency of our reported gains in performance.
>
> ---
>
> **W1:** We will revise the writing to clarify how previous trajectories are retrieved and to better explain our ablative study in Table 9. Briefly, Table 9 examines the impact of injecting random in-context examples versus cosine-similarity-based examples. In phase 1, successful trajectories and the embedding of the problem's docstring are added to the memory bank to facilitate retrieval. In phase 2, when re-attempting failed problems, we generate an embedding for the docstring and retrieve the k most similar trajectories from the memory-bank to inject as in-context examples. The results indicate that using similar-style examples boosts performance over random ones, aligning with recent findings in [1].
>
> ----
>
> **W2:** We will revise the plots to place both embeddings in the same space, as suggested, to improve clarity. Additionally, we introduce a new metric to quantify the diversity of generated reflections. This metric calculates the average pairwise cosine similarity of self-reflections for each problem within a dataset; we report both the mean and standard deviation. We use the ‘all-Mini-LM-v6’ model from SentenceTransformers to generate embeddings for these reflections. Diversity scores are presented for HumanEval and LeetCodeHardGym across two different LLMs, with lower similarity scores indicating greater diversity.
>
> |               | **HumanEval**        |        |
> | ------------- | ---------------- | ------ |
> | **Llama-3.1-70B** | **Similarity Score** | **Pass@1** |
> | **LATS**          | 0.65 (0.41)      | 84.76  |
> | **Reflexion**     | 0.83 (0.09)      | 87.2   |
> | **DoT**           | **0.48 (0.11)**      | **89.63**  |
> | **DoT-Bank**      | **0.48 (0.12)**      | **93.29**  |
> |               |                  |        |
> |               |                  |        |
> |               | **LeetCodeHardGym**  |        |
> | **Sonnet3.5**     | **Similarity Score** | **Pass@1** |
> | **LATS**          | 0.70 (0.29)      | 42.5   |
> | **Reflexion**     | 0.83 (0.13)      | 42.5   |
> | **DoT**           | **0.62 (0.13)**      | **50**     |
> | **DoT-Bank**      | **0.61 (0.10)**      | **52.5**   |
>
> ---
>
> **W3:** We agree with the reviewer. We will reformat and reposition the tables to enhance readability and clarity.
>
> ----
>
> **Q1:** Tree-of-Thoughts and LATS use LLMs as evaluators, where the LLM is prompted to generate a numeric score, as noted by the reviewer.
>
>
> **Q2:** Yes, the K most similar trajectories are injected as in-context examples specifically for generating function implementations, not reflections. As stated in the paper, the task-agnostic memory bank aims to enhance generation quality.
>
>
> ----
>
> References
>
> [1] Metacognitive Capabilities of LLMs: An Exploration in Mathematical Problem Solving, Didolkar. et. al, NeurIPS 2024

---

> > ### Comment · Reviewer_vf4f · 2024-11-14
> >
> > I thank the authors for their response. The suggested changes all look sensible to me and I commend the inclusion of average cosine-similarities as a measure of diversity.

---

### Official Review · Reviewer_vtDD · 2024-11-02

**Soundness:** 2
**Presentation:** 2
**Contribution:** 2
**Rating:** 6
**Confidence:** 4

**Summary:**

This paper introduces DoT (Diversity of Thoughts), a novel framework for improving language model agents's iterative reasoning tasks by addressing two key limitations in existing approaches: 1: redundant reflections that lead to inefficient exploration of the decision space, and 2:the inability to leverage insights across different tasks. The framework consists of a diverse-reflections model that reduces redundancy in reasoning paths and a task-agnostic memory bank that enables knowledge transfer between tasks. Through extensive experiments on programming benchmarks (HumanEval, MBPP, and LeetCodeHardGym) using various language models, DoT demonstrates up to 10% improvement in Pass@1 while maintaining cost-effectiveness compared to existing methods like LATS, and when integrated with Tree of Thoughts (ToT), its diverse reflection module improved performance by 13% on the Game of 24 benchmark.

**Strengths:**

The paper identifies and addresses concrete limitations in existing approaches (redundant reflections and isolated task solving) with a practical solution that shows consistent improvements across different models and benchmarks. The proposed modifications are simple to implement yet effective.

**Weaknesses:**

1:The paper's two main components (Diverse-Reflection and Memory Bank) appear to be independent modules without a strong theoretical connection or necessity to be combined. This suggests the work is more of an engineering effort combining separate improvements rather than a cohesive theoretical advancement. The lack of comprehensive ablation studies makes it unclear which component truly drives the performance gains.

2: The paper's positioning around efficiency improvement over LATS is problematic because LATS and similar resource-intensive methods primarily aim to maximize accuracy, not efficiency. The experimental validation has several limitations: improvements are incremental (2-10%), test sets are relatively small, and no statistical significance testing is reported to validate that improvements are meaningful rather than random variation.

3: The paper lacks detailed analysis of why and how diversity in reflections leads to better results. There's insufficient discussion about the reliability of one-shot sampling for generating diverse reflections and potential failure modes. Also, key technical details about the memory bank implementation and retrieval mechanism are not fully explained, including specifics about how "relevant trajectories" are defined and retrieved and how the diversity in reflection generation is ensured; all of these make reproduction challenging.

4: The evaluation focuses mainly on programming tasks with visible or synthetic unit tests, limiting the method's demonstrated applicability to domains where clear evaluation metrics are available. This leaves questions about generalizability to broader reasoning tasks.

**Questions:**

How is cost calculated for different model types (especially for Llama models where costs are in GPU hours rather than API calls)? Why is the cost difference between iterative and one-shot sampling surprisingly small in Table 10?

What specific mechanism is used for retrieving "relevant trajectories" from the memory bank? How is relevance defined and measured?

While the method reduces LLM inference calls by avoiding redundancy, does it require more tokens per call due to longer context? Can you provide a quantitative analysis of average input/output token lengths compared to baselines?

What's the effect of varying k in k diverse-reflections? How reliable is the one-shot prompt approach, and what's the sensitivity to the number of retrieved trajectories?

---

> ### Author Response · Authors · 2024-11-15
> **Response 1/3**
>
> We thank the reviewer for their valuable feedback. We address the raised weaknesses and questions below.
>
> **W1.** We thank the reviewer for raising this point. We will clarify in the paper that the ablation you suggest is already captured by the difference between our two variants, DoT and DoT-Bank, which isolates the contributions of each module. Specifically:
>
> 1. **DoT** utilizes the Diverse-Reflections module and achieves an average gain of **~4.5%** over the best competing method across all datasets and LLMs.
>
> 2. **DoT-Bank** incorporates the task-agnostic memory constructed during Phase-1 and further improves performance by an average of **~4.4%**.
>
> These results demonstrate that both components contribute independently to the performance gains, with their combination offering the highest improvements.
>
> The key innovation lies in identifying and addressing the issue of repetition in naive reflections, which we quantify in the table below. Repetitive prompting fails to improve model performance, while our Diverse-Reflections method demonstrably generates more varied outputs, as evidenced by lower pairwise cosine similarity scores (quantified using the ```all-Mini-LM-v6 model``` from SentenceTransformers). These metrics are provided for HumanEval and LeetCodeHardGym across two LLMs and will be included in the final paper.
>
> |               | **HumanEval**        |        |
> | ------------- | ---------------- | ------ |
> | **Llama-3.1-70B** | **Similarity Score** | **Pass@1** |
> | **LATS**          | 0.65 (0.41)      | 84.76  |
> | **Reflexion**     | 0.83 (0.09)      | 87.2   |
> | **DoT**           | **0.48 (0.11)**      | **89.63**  |
> | **DoT-Bank**      | **0.48 (0.12)**      | **93.29**  |
> |               |                  |        |
> |               |                  |        |
> |               | **LeetCodeHardGym**  |        |
> | **Sonnet3.5**     | **Similarity Score** | **Pass@1** |
> | **LATS**          | 0.70 (0.29)      | 42.5   |
> | **Reflexion**     | 0.83 (0.13)      | 42.5   |
> | **DoT**           | **0.62 (0.13)**      | **50**     |
> | **DoT-Bank**      | **0.61 (0.10)**      | **52.5**   |
>
> We would also like to point out that the use of memory banks is now a seemingly standard best practice that *ANY* iterative prompting approach should incorporate (we compare and contrast again some recent works that use memory-banks in Section-4); this is our main reason for including it in our work. While **DoT-Bank** shows higher performance, **DoT** itself surpasses prior state-of-the-art methods, underscoring the independent value of both components.
>
> ----
>
> **W2.** We thank the reviewer for raising these concerns. First, we would like to clarify that the primary focus of our method is also on **maximizing accuracy**, with efficiency as an ancillary benefit. We highlighted the efficiency comparison with LATS because its high cost is tied to its inherent statistical inefficiency, caused by redundant reflections—a key issue our method directly addresses. The Diverse-Reflections module in DoT demonstrates that effective results can be achieved without resorting to complex, resource-intensive methods, offering both efficacy and efficiency.
>
> Regarding the experimental setup, we have followed the same methodology as foundational papers in this area, such as Reflexion, Tree-of-Thoughts, and LATS. While prior works did not report statistical significance testing, we agree that it is essential to validate our results further. As a first step, please see the new table below that includes Pass@1 averaged over 3 runs for HumanEval and LeetCodeHardGym datasets using Sonnet-3.5, which we will incorporate into the final version of the paper.
>
> **Updated Results (Pass@1) with Statistical Significance** (LLM: Sonnet-3.5, averaged over 3 runs)
>
> | **Model** | **HumanEval** | **LeetCodeHardGym** |
> |-----------|---------------|---------------------|
> | Base | 85.57±0.94 | 40.83±2.89 |
> | Reflexion | 88.21±0.35 | 41.67±1.44 |
> | LATS | 89.16±0.68 | 42.50±2.50 |
> | **DoT** | **91.46±0.61** | **50.00±2.50** |
> | **DoT-Bank** | **94.10±0.35** | **53.33±1.44** |
> -----
>
> On the claim that our gains are incremental, we respectfully disagree. Reflexion introduced LeetCodeHardGym as a benchmark for challenging tasks, and our method (DoT) achieves **up to 12.5% improvement** over Reflexion and LATS across various LLMs. Furthermore, on average:
> 1. **DoT** solves 7 more problems on HumanEval, 18 on MBPP, and 2 on LeetCodeHardGym compared to prior best methods.
> 2. **DoT-Bank** solves 14 more problems on HumanEval, 36 on MBPP, and 4 on LeetCodeHardGym compared to previous best methods.
>
> These improvements are not tuned for specific LLMs or datasets, reinforcing that the gains are robust and not attributable to random variation. The consistent performance across datasets and LLMs provides additional evidence of the generality and significance of our approach.

---

> > ### Author Response · Authors · 2024-11-15
> > **Response 2/3**
> >
> > **W3.A. Impact of Diverse Reflections on Performance.**
> > Our analysis revealed that prior methods, such as LATS and Reflexion, often generate repetitive self-reflections, limiting their effectiveness. In contrast, we hypothesized that generating diverse reflections allows for a broader exploration of the decision space, thereby improving performance. This hypothesis is supported by consistent gains across various LLMs and datasets, as detailed in our results. Additionally, we introduced a quantitative diversity metric (see W1) to measure the effectiveness of our approach, showing a clear reduction in repetitive reflections and an increase in semantic diversity.
> >
> > **W3.B. Reliability of One-Shot Sampling:**
> > A concurrent study [1] corroborates our approach, showing that one-shot sampling can produce semantically diverse outputs. We will incorporate this study into the discussion and expand the diversity metrics across all relevant tables in the final version of the paper.
> >
> > **W3.C. Technical Details of the Memory Bank Implementation:**
> > To enhance reproducibility, we provide the following details about the memory bank:
> > The memory bank is implemented as a hash-map: {unique-task-id: complete-task-trajectory}.
> > * Phase 1: Successful trajectories and the embedding of the task’s docstring are stored in the memory bank.
> > * Phase 2: When re-attempting failed problems, we generate an embedding for the task’s docstring and retrieve the k most similar trajectories using cosine similarity.
> > * We use ```Cohere-Embed-V3-English``` for generating embeddings.
> > The k closest trajectories, based on cosine similarity, are considered "relevant" and used as in-context examples for the actor agent.
> >
> > We have provided the full implementation in the supplementary materials, including all details necessary to reproduce the memory bank and retrieval mechanism.
> >
> >
> > **W4.** We appreciate the reviewer’s observation. As highlighted in the Reflexion paper, programming tasks provide a unique setting for evaluating iterative frameworks due to the availability of synthetic or visible test cases, which enable objective performance assessment. Accordingly, we conducted an extensive evaluation across three programming benchmarks and several LLMs, as well as testing our diverse reflections approach on **Game-of-24**, a reasoning task highlighted in Tree-of-Thoughts.
> >
> > We agree that testing on a broader range of reasoning tasks is important for assessing generalizability beyond programming contexts. In the final version of the paper, we will expand our evaluation to include additional reasoning tasks to provide a more comprehensive validation of our approach. If the reviewer has specific reasoning tasks in mind, we would greatly appreciate the suggestions.
> >
> > Additionally, we will update the paper with detailed implementation notes for the memory bank and other relevant aspects to enhance clarity and reproducibility. Thank you again for your valuable feedback.
> >
> > ----
> >
> > **Q1.** We would like to clarify that by “cost” we refer solely to the dollar cost incurred by running the LLM APIs (dictated by total API invocations, input and output tokens).
> > * For GPT based models, we use the official API pricing from: https://openai.com/api/pricing/
> > * For Claude models, we use the official API pricing from: https://www.anthropic.com/pricing
> > * For Llama models, we use Amazon’s BedRock API service and pricing from: https://aws.amazon.com/bedrock/pricing/
> >
> > The cost difference between iterative and one-shot sampling in Table 10 is small because, while iterative sampling involves appending all previously generated reflections to the input context (causing the input token count to grow), the output token count remains roughly the same in both methods. For both approaches, we generate three diverse reflections, keeping the output size comparable.
> >
> > However, the cost difference becomes more pronounced as the API cost of the underlying LLM increases. For example:
> > * **GPT-4o-mini:** Input tokens cost **$0.15/1M**, and output tokens cost **$0.60/1M**. This results in a cost difference of **$0.02** between iterative and one-shot sampling.
> > * **GPT-o1**: Input tokens cost **$15/1M**, and output tokens cost **$60/1M**. Here, the cost difference increases significantly to **$10.62**.
> >
> > This trend highlights that while the cost impact may appear minimal for lower-cost models, it scales significantly with higher-cost APIs, making the choice between iterative and one-shot sampling more critical in such scenarios.
> >
> > **Q2.** As discussed in our response to W3, we embed the docstring for each problem to facilitate retrieval. We rely on cosine-similarity metric and select ‘j’ most similar trajectories (deemed as relevant trajectories) and inject them into context for the actor agent.

---

> > > ### Author Response · Authors · 2024-11-15
> > > **Response 3/3**
> > >
> > > **Q3.** The cost of an LLM inference call is a function of input and output tokens as laid out previously. We include a table below for reference to shed light on the token analysis for DoT variants and baselines.
> > >
> > > | HumanEval-Sonnet-3.5 | Avg Input Tokens | Avg Output Tokens | Avg Cost |
> > > | -------------------- | ---------------- | ----------------- | -------- |
> > > | **Base**                 | 223              | 114               | $0.0024  |
> > > | **Reflexion**            | 702              | 221               | $0.0054  |
> > > | **LATS**                 | 9370             | 1230              | $0.0466  |
> > > | **DoT**                  | 904              | 282               | $0.0070  |
> > > | **DoT-Bank**             | 1418             | 395               | $0.0102  |
> > >
> > > These metrics (per task) are also available for all our experiments in the log files we provided as part of supplementary material.
> > >
> > > **Q4.** Setting k=1 in k diverse-reflections, would make M_dsr similar to M_sr. We set k=3 to strike a balance between exploration and cost – we will ablate on other choices of k and report its impact on performance and cost.
> > >
> > > A recent concurrent study [1] supports our approach, demonstrating that one-shot sampling can produce semantically diverse outputs. We also quantified diversity in our above response, validating DoT can reduce redundancies in reflections.
> > >
> > > In Table 10, we ablate on the impact of the number of retrieved trajectories on Pass@1. We observe a bell-shaped behavior: performance improves as the number of injected trajectories increases, up to a certain threshold. Beyond this point, adding more examples degrades performance, as the task context gets overshadowed by excessive examples.
> > >
> > > ----
> > >
> > > We would be glad to address any remaining concerns the reviewer may have or provide further clarifications. Additionally, we will update our paper to incorporate all new discussions and clarifications.
> > >
> > > ----
> > >
> > > References
> > >
> > > [1] How Far Can We Extract Diverse Perspectives from Large Language Models? Hayati. et. al.,  EMNLP 2024

---

> ### Comment · Reviewer_vtDD · 2024-11-18
> **reply**
>
> I appreciate the author's effort to address my concerns, particularly their detailed response and inclusion of new experiments. However, two main concerns remain.
>
> First, regarding its practical usability: while the authors demonstrate improvements over foundational algorithms like LATS and reflexion in both accuracy and efficiency, to substantiate claims of SOTA performance in Agent framework design in coding tasks, which are the main focus of this paper, they should include comparisons with current SOTAs in the field [1, 2]. Additionally, the discussion about time efficiency would benefit from addressing computational overhead related to embedding vectors and similarity calculations. Given the significant engineering complexity of this work, at least some short sample anonymized code release would also be very helpful for reproducing the work.
>
> Second, concerning the paper's core contribution about diverse thought helping LLM reasoning: this idea, while valuable, has been explored in several recent works [3, 4], and the current paper lacks either theoretical foundations or social science perspectives to differentiate its contribution from existing literature.
>
> Nevertheless, I commend the authors' efforts in addressing my initial concerns, their comprehensive experiments and ablation studies, and the clear writing style. Given these considerations, I am willing to raise my score to 6, though I still remain neutral on the accept/reject decision and defer to other reviewers and ACs.
>
> References:
> [1] Zhong, L., Wang, Z., & Shang, J. (2024). Debug like a Human: A Large Language Model Debugger via Verifying Runtime Execution Step by Step. Findings of ACL 2024, 851-870.
>
> [2] Huang, D., Zhang, J. M., Luck, M., Bu, Q., Qing, Y., & Cui, H. (2024). AgentCoder: Multi-Agent-based Code Generation with Iterative Testing and Optimisation. arXiv preprint arXiv:2312.13010.
>
> [3] Hayati et al. (2024). How Far Can We Extract Diverse Perspectives from Large Language Models? EMNLP 2024.
>
> [4] Muscato, B., Mala, C. S., Manerba, M. M., Gezici, G., & Giannotti, F. (2024). An Overview of Recent Approaches to Enable Diversity in Large Language Models through Aligning with Human Perspectives. Proceedings of the 3rd Workshop on Perspectivist Approaches to NLP, 49-55.

---

> ### Author Response · Authors · 2024-11-18
>
> We sincerely thank the reviewer for their constructive feedback and appreciate their willingness to raise the score. Below, we address the two main concerns raised:
>
> **Comparisons with Recent SOTA and Practical Usability:**
> We appreciate the reviewer pointing us to recent SOTA works in agentic frameworks for coding tasks. In response, we conducted additional experiments comparing DoT variants against LDB [1], an accepted and well-established work, on the HumanEval dataset using GPT-4o and GPT-4o-mini. We report Pass@1 results (averaged over three runs with standard deviations) below.
> Notably, DoT variants demonstrate superior performance over LDB. We hypothesize that integrating DoT-generated seed programs into LDB’s framework may further enhance its performance. This highlights the complementary nature of our contributions.
>
> | **Method**   | **GPT-4o-mini**  | **GPT-4o**       |
> | -------- | ------------ | ------------ |
> | **LDB [1]**      | 89.02 ± 0.61 | 93.90 ± 0.61 |
> | **DoT**      | **92.88 ± 0.93** | **95.95 ± 0.70** |
> | **DoT-Bank** | **95.93 ± 0.35** | **96.54 ± 0.35** |
>
>  Regarding computational overhead, as mentioned in our prior response, embedding similarity computations are negligible for datasets like HumanEval and MBPP, especially when embeddings are generated using efficient APIs. We will include a detailed discussion of this in the final version.
>
> **Additionally, we would like to clarify that all code and log files have already been released as part of the supplementary material in the initial submission to facilitate reproducibility.**
>
> -----
>
> **Contribution on Diverse Reflections Enhancing Reasoning:** We note that [3] was recently accepted shortly before the ICLR25 abstract deadline, and we appreciate the opportunity to incorporate its insights in our updated PDF. We acknowledge that [3, 4] have explored one-shot sampling to produce semantically diverse outputs. However, these works do not specifically investigate how diversity aids reasoning—a key focus and contribution of our study.
>
> Our work identifies principled shortcomings in prior iterative frameworks like Reflexion and LATS and demonstrates how diversity in reflections can address these issues. This contribution is novel and, to the best of our knowledge, represents the first exploration of diversity-enhanced reasoning in this context. While [3, 4] align with our findings, they provide orthogonal insights rather than overlapping contributions. Also, our updated PDF and previous response acknowledges [3].
>
> -----
>
> We hope these clarifications address the reviewer’s concerns and reinforce the novelty and practical value of our work. We appreciate the detailed review and look forward to further feedback.

---

> > ### Comment · Reviewer_vtDD · 2024-11-18
> > **re**
> >
> > I think sentence transformer embeddings' efficiency (and accuracy) still needs more thorough analysis, particularly when using a high k value for diverse reflections.
> >
> > Regarding reproducibility, I am specifically referring to the additional experiments you conducted. While your submitted code utilizes API calls (including for LLaMA), which makes large-scale reproduction challenging, it would be valuable to see the detailed implementation of these experiments, given their significance to your paper's conclusions.
> >
> > As for the question of novelty in Diverse Reflections, I have made my point and will defer to the AC's judgment on this matter.
> >
> > Thanks.

---

> > > ### Author Response · Authors · 2024-11-25
> > >
> > > We thank the reviewer for their feedback and follow-up questions. Below, we address the concerns and clarify potential misunderstandings:
> > >
> > > 1. **Efficiency of Sentence Transformer Embeddings**
> > >    The Sentence Transformer embeddings (`all-mini-LM-v6`) were used only for post-hoc analysis, specifically to compute the diversity metric of generated reflections, and are not part of the proposed method itself. Thus, concerns about their efficiency are unrelated to our core contributions.
> > >
> > >
> > >    To strengthen our analysis, we recalculated the diversity metric using OpenAI’s `text-embedding-3-large` model [1], which achieves 64.6% on the MTEB benchmark, demonstrating its competitiveness. The new results are presented in the table below -- DoT variants generate more diverse (less redundant) reflections.
> > >
> > > ---
> > > |               |       HumanEval                                 |               |
> > > | ------------- | ----------------------------------------- | ------------- |
> > > | **Llama-3.1-70B** | **Similarity Score (text-embedding-3-large)** | **Pass@1**        |
> > > | LATS          | 0.69 (0.36)                               | 84.76         |
> > > | Reflexion     | 0.82 (0.09)                               | 87.2          |
> > > | **DoT**           | **0.50 (0.09)**                               | **89.63**         |
> > > | **DoT-Bank**      | **0.50 (0.10)**                               | **93.29** |
> > >
> > >
> > >    Additionally, we chose \( k = 3 \) diverse reflections in our experiments to balance cost and performance. Following the reviewer’s suggestion, we will include a detailed analysis of how cost and performance vary with \( k \) in the final version.
> > >
> > >
> > > 2. **Large-Scale Reproduction**
> > >    We respectfully disagree with the concern about reproduction. Like other baselines, our method relies on APIs, which are significantly more cost-effective for reproducing experiments compared to procuring GPUs to run large-scale models such as LLaMA-3.
> > >
> > >
> > > 3. **Core Contribution and Relation to Prior Work**
> > >    We revisited the related work [2, 3] the reviewer suggested in greater detail, and note that these works primarily focus on increasing diversity in LLM outputs through socio-demographic prompting, criteria-based prompting, or RLHF to align with human perspectives. In contrast, our work fundamentally differs by leveraging diverse self-reflections to enhance reasoning abilities. Our focus on iterative diversity-driven refinement for improving reasoning is distinct from the broader ethical and social considerations explored in [2, 3].
> > >
> > > We hope these clarifications address the reviewer’s concerns. We deeply appreciate their engagement and valuable suggestions during this discussion phase.
> > >
> > > -----
> > >
> > > References
> > >
> > > [1] https://openai.com/index/new-embedding-models-and-api-updates/
> > >
> > > [2] Hayati et al. (2024). How Far Can We Extract Diverse Perspectives from Large Language Models? EMNLP 2024.
> > >
> > > [3] Muscato, B., Mala, C. S., Manerba, M. M., Gezici, G., & Giannotti, F. (2024). An Overview of Recent Approaches to Enable Diversity in Large Language Models through Aligning with Human Perspectives. Proceedings of the 3rd Workshop on Perspectivist Approaches to NLP, 49-55.

---

### Official Review · Reviewer_RxnW · 2024-11-02

**Soundness:** 2
**Presentation:** 2
**Contribution:** 2
**Rating:** 5
**Confidence:** 4

**Summary:**

This paper proposes Diversity of Thought (DoT), a method that enhances LLM reasoning through diverse reflections. The paper first identifies two limitations of existing methods: (1) limited exploration due to repetitive reflections, and (2) inability to reuse knowledge from similar tasks. To address these, DoT reduces repetitive reflections and incorporates a task-agnostic memory. Experiments on three programming benchmarks and the Game of 24 demonstrate improvements of DoT compared to baselines.

**Strengths:**

- The paper examines tasks requiring complex reasoning and proposes solutions to enhance LLMs' reasoning capabilities. Experiments show that DoT and DoT-bank outperform baselines across various benchmarks.
- The paper provides some analysis, e.g. quantifying the redundancy of reflections generated using existing methods, to motivate the proposed method.

**Weaknesses:**

- The proposed method is motivated by two observed limitations of existing methods: lack of exploration and inability to leverage cross-task knowledge. While the first limitation is analyzed and discussed in Table 1, the second lacks supporting analysis. Therefore, adding a task-agnostic memory component to DoT appears somewhat disconnected from the paper’s discussion. Although intuitively it makes sense to reuse insights from previously solved tasks, whether it is useful in practice remains unclear. The authors need to provide more evidence to justify this proposed solution in order to create a more coherent story for the paper.
- For the Reflexion baseline, is the number of reflections set to 1? If so, this may not be a fair comparison, as DoT uses many more reflections. A better comparison would be to Reflexion with k sampled reflections. This would more accurately indicate whether the proposed diverse reflection generation method (one-shot or iterative) is effective.
- DoT-bank requires building a memory bank, and its size scales with the number of failed tasks using DoT based on Algorithm 1. Therefore, the reported cost of DoT-Bank is misleading, as it does not include the cost of building the memory bank. Also, for the DoT-Bank results in all tables, is the memory bank used for every task or only for failed tasks?
- The choice of models used for different tasks seems arbitrary. Why are different models used across tasks? For example, why does HumanEval use Sonnet 3.5 and Llama-3.1, while LeetCodeHardGym only uses Sonnet 3.5? Additionally, why does GPT-3.5 have a different setting in Table 5 compared to other models in Table 4?
- For incorporating DoT into ToT, is the only difference that ToT samples k thoughts in parallel, while ToT+Diversity samples k thoughts in one shot?

**Questions:**

- L700 Figure reference is missing
- The paper uses Reflexion as a key baseline, but other similar self-reflection methods, e.g. [1], should also be cited.
- Several citation formats are not proper, e.g. some \citet, and \citep are misused, e..g L41, L105, L350, etc.
- How is the cost of the Llama-3.1 model calculated?

[1] Aman Madaan, Niket Tandon, Prakhar Gupta, Skyler Hallinan, Luyu Gao, Sarah Wiegreffe, Uri Alon, Nouha Dziri, Shrimai Prabhumoye, Yiming Yang, Shashank Gupta, Bodhisattwa Prasad Majumder, Katherine Hermann, Sean Welleck, Amir Yazdanbakhsh, Peter Clark. Self-Refine: Iterative Refinement with Self-Feedback

---

> ### Author Response · Authors · 2024-11-14
> **Response 1/2**
>
> We thank the reviewer for their valuable feedback.
>
> **W1:** We disagree with the reviewer’s comment regarding the practicality of reusing insights from previously solved tasks. Our empirical results underscore the practical value of DoT-Bank, showing a consistent **2-5%** performance improvement over the standard DoT approach—validating that leveraging past insights enhances task-solving effectiveness.
>
> A recent concurrent study [1] demonstrates that Large Language Models (LLMs) can effectively use metacognitive abilities to leverage previous insights when approaching new tasks, leading to substantial performance improvements. Specifically, they show that curating in-context examples based on skill-exemplars—i.e., insights from related tasks—enhances performance. However, their approach relies on a predefined set of exemplars, requiring additional training data, which may limit adaptability across varied tasks.
>
> Our proposed task-agnostic memory bank variant, "DoT-Bank," corroborates this observation but innovates by eliminating the need for explicit training data. Instead, DoT-Bank dynamically builds a memory bank during problem-solving, effectively capturing useful knowledge from previously solved tasks. This enables DoT-Bank to leverage cross-task insights without task-specific exemplars.
>
> ---
>
> **W2:** As stated in Line 300, we use the recommended hyperparameters suggested by the authors of the respective baselines. Specifically, for Reflexion, we set max-iterations k=3 for HumanEval and MBPP, and k=5 for LeetCodeHardGym. For LATS, we use k=8 across all three datasets. For DoT and DoT-Bank, we set k=3 for HumanEval and MBPP, and k=5 for LeetCodeHardGym.
>
> To further highlight DoT's effectiveness, we vary k in Reflexion and report both Pass@1 and associated costs on the HumanEval dataset with **Sonnet-3.5**. The results indicate that even with k=10, Reflexion lags behind **DoT** and **DoT-Bank**, which achieve Pass@1 scores of **91.46 ($1.14)** and **93.90 ($1.67)**, respectively—demonstrating that DoT variants are both more effective and cost-efficient.
>
> | HumanEval | K=3           | K=6           | K=8           | K=10          |
> | --------- | ------------- | ------------- | ------------- | ------------- |
> | Reflexion | 88.41 ($0.89) | 89.02 ($1.36) | 89.63 ($1.64) | 89.63 ($1.87) |
>
> If the reviewer is referring to sampling k-reflections per iteration, this approach is already implemented in LATS, which additionally uses an MCTS mechanism to select a potential solution (see Section 5.2 paragraph 2 in LATS). Our empirical results show that DoT outperforms LATS, and in the few cases of similar performance, DoT remains more cost-effective.
>
> ---
>
> **W3:** We clarify that by “cost”, we refer solely to the dollar cost incurred by running the LLM APIs. The memory bank is implemented as a hash map, which does not contribute to this cost.
> It appears there may have been a misunderstanding regarding our algorithm. The memory bank's size increases with each new solved task (Algorithm 1, Lines 17 and 35). DoT-Bank includes both phases of the algorithm: phase 1, which builds the memory bank and tags failed problems on visible or synthetic test cases, and phase 2, which reattempts these tagged problems using the memory bank. For all DoT-Bank experiments, both phases are run. This is why, in all relevant tables, DoT-Bank's cost is higher than DoT’s—reflecting the additional cost of reattempting failed problems with extra context tokens and the API cost for generating embeddings.
>
> The memory bank is only applied to failed tasks, as shown on Line 32 of Algorithm 1.
>
> ---
>
> **W4:** We use the same set of models for HumanEval and MBPP experiments. For the LeetCodeHardGym benchmark, consistent with Reflexion, we utilize powerful LLMs as base models, specifically GPT-o1 and Sonnet-3.5 in the main paper, and GPT-4o/4o-mini in Table 11 of Appendix.
>
> Table 5 highlights a bug in the LATS implementation that led to inflated performance results on some datasets. To illustrate this, we use the same setting from LATS with synthetic test cases. All baseline results in Table 5 are taken directly from the LATS paper, alongside our corrected results for LATS and DoT.
>
> Lastly, we emphasize that, unlike previous works, our study evaluates a broad range of LLMs, demonstrating consistent performance gains across different model types.
>
> ---
>
> **W5:** For ToT + DoT, we use one-shot-sampling along with the prompt mentioned in Figure 8. A recent concurrent study [2] supports our approach, demonstrating that one-shot sampling can produce semantically diverse outputs.
>
> ---
>
> We updated our paper to make these points clear in addition to adding new experiments and discussions.
>
> ---
> References:
>
> [1] Metacognitive Capabilities of LLMs: An Exploration in Mathematical Problem Solving, Didolkar. et. al, NeurIPS 2024
>
> [2] How Far Can We Extract Diverse Perspectives from Large Language Models? Hayati. et. al.,  EMNLP 2024

---

> > ### Author Response · Authors · 2024-11-14
> > **Response 2/2**
> >
> > **Q1:** Thank you for pointing this out. The missing reference in line 700 corresponds to Figure 7.
> >
> > **Q2:** We will cite Self-Refine and other related Reflexion based methods in our final version.
> >
> > **Q3:** We apologize for our oversight on inconsistent citations. We will fix them in our final version.
> >
> > **Q4:** We use Amazon’s BedRock service to access Llama models APIs. The corresponding API pricing can be found here: https://aws.amazon.com/bedrock/pricing/

---

> ### Author Response · Authors · 2024-11-20
> **Following Up with the Reviewer**
>
> Thank you again for your review. As the discussion period is concluding soon, we would like to follow up to ensure we have sufficiently addressed your concerns and to kindly ask if you might consider reassessing your evaluation.
>
> To summarize our updates:
>
> 1. We added a connecting paragraph in Section I, citing a relevant recently accepted work to strengthen support for our memory-bank (insights from previous tasks are helpful) and improve the narrative coherence.
>
> 2. We included a new study in Appendix A.2.5 to show that naively increasing total generated reflections in Reflexion has minimal impact on performance. Appendix A.2.1 shows that our reported gains are statistically significant. We hope that these new experiments and clarifications make our contributions clear.
>
> 3. We additionally included a diversity metric in updated Section 3.3.1 to show that DoT can reduce redundant reflections compared to baselines.
>
> We hope these additions clarify our contributions and address your concerns. We would be happy to answer any follow-up questions and address any remaining concerns.

---

> > ### Comment · Reviewer_RxnW · 2024-11-23
> > **Response to the Authors**
> >
> > Thank you to the authors for the response! The response has addressed most of my concerns, and I have adjusted my score accordingly. For W3, I was referring to the first phase of the DoT-bank, which scales with the number of examples in the dataset. In other words, if I want to solve one specific problem that DoT cannot solve, using the DoT-bank requires running DoT on the entire dataset to build the memory bank. However, other algorithms can simply solve this one problem, which might be much more cost-efficient. This might not be a significant issue for evaluation purposes (as the entire dataset needs to be processed anyway), but it could be a problem in real applications.

---

> > > ### Author Response · Authors · 2024-11-23
> > >
> > > We thank the reviewer for their response and for updating the score. We would like to clarify a potential misunderstanding regarding DoT-Bank. The first phase of the DoT algorithm (Phase-1) builds a task-agnostic memory bank for the dataset. **This memory bank can be reused across tasks, meaning that for any specific problem DoT cannot solve, only Phase-2 of the DoT-Bank algorithm needs to be executed.** This makes DoT-Bank cost-efficient for reattempting failed problems, as the initial Phase-1 step is not repeated.
> > >
> > > To further highlight the effectiveness of DoT, we compared DoT variants against LDB [1], a state-of-the-art agentic approach for coding tasks. Below, we present results averaged over three runs with standard deviations. Our experiments demonstrate that DoT achieves superior performance in these scenarios.
> > >
> > > | **Method**   | **GPT-4o-mini**  | **GPT-4o**       |
> > > | -------- | ------------ | ------------ |
> > > | **LDB**      | 89.02 ± 0.61 | 93.90 ± 0.61 |
> > > | **DoT**      | **92.88 ± 0.93** | **95.95 ± 0.70** |
> > > | **DoT-Bank** | **95.93 ± 0.35** | **96.54 ± 0.35** |
> > >
> > > We hope this resolves the concern, and we are happy to address any remaining questions.

---

> > > > ### Author Response · Authors · 2024-11-27
> > > > **A Gentle Reminder to Reviewer RxnW**
> > > >
> > > > Dear Reviewer RxnW,
> > > >
> > > > Thank you for your valuable feedback. To better demonstrate the cross-task knowledge transfer capability of our task-agnostic memory bank in DoT-Bank, we conducted an additional study. Specifically, we utilized the memory bank generated by DoT (Phase 1 of our algorithm) on the MBPP dataset to evaluate DoT-Bank on the HumanEval dataset. This variant, referred to as DoT-Bank-MBPP, is summarized in the table below. The results demonstrate that DoT-Bank effectively leverages insights from related datasets, underscoring its flexibility and cost-efficiency.
> > > >
> > > > |               | GPT-4o-mini |       |
> > > > | ------------- | ----------- | ----- |
> > > > | **Method**        | **Pass@1**      | **Cost**  |
> > > > | Base          | 87.2        | $0.02 |
> > > > | Reflexion     | 90.85       | $0.04 |
> > > > | LATS          | 90.24       | $0.24 |
> > > > | **DoT**           | **92.07**       | **$0.05** |
> > > > | **DoT-Bank**      | **95.73**       | **$0.07** |
> > > > | **DoT-Bank-MBPP** | **94.52**       | **$0.06** |
> > > >
> > > >
> > > > We kindly request the opportunity to address any remaining concerns you may have.

---

> ### Author Response · Authors · 2024-12-02
> **Follow-up**
>
> Dear Reviewer RxnW,
>
> We hope our response has adequately addressed your concerns. As the discussion period nears its conclusion, we kindly ask if you have any remaining questions or if you could reassess our paper based on the clarifications provided.
>
> Thank you for your time and consideration.

---

> > ### Comment · Reviewer_RxnW · 2024-12-03
> >
> > Thank you to the authors for providing additional results. I think the cross-task knowledge transfer results are helpful and should be included. However, I still share a similar concern with the other reviewer that improving the diversity of output has been widely explored. Therefore, I will maintain my original rating: from 3 to 5.

---

> > > ### Author Response · Authors · 2024-12-03
> > >
> > > We thank the reviewer for their response and are encouraged that they found our cross-task knowledge transfer results valuable. We respectfully submit that while diversity in outputs has been broadly explored, our work is, **to the best of our knowledge, the first to explicitly employ diverse reflections to enhance reasoning abilities in this manner.** Without specific references to closely related prior work, we believe our approach represents a novel and impactful contribution to the field. We hope this distinction will help the reviewer reconsider their assessment.

---

### Official Review · Reviewer_7Wju · 2024-11-06

**Soundness:** 2
**Presentation:** 3
**Contribution:** 2
**Rating:** 5
**Confidence:** 3

**Summary:**

Motivated by the observation of repetitive reflections, this paper proposes a method to improve the diversity of generated reasoning by incorporating task-agnostic memory and diversity of thoughts (DoT) prompting (on top of reflexion). The experiment results show clear improvement with various LLMs.

**Strengths:**

1. The method is well-motivated.
2. I appreciate the analysis and results in the Introduction to showcase the repetitive reasoning generation with existing methods, i.e., LATS, which is insightful and provides evidence.
3. The paper is well-written and easy to follow.
4. The results seem promising.

**Weaknesses:**

1. I am still unsure if diverse reflections can be guaranteed if prompting the Mdr agent with "Ensure your reflections are accurate and leverage previous reflections to avoid repetition. Aim for diversity in your explanations while prioritising the correctness of your hints.". Can you provide empirical evidence or analysis demonstrating how your prompting approach leads to diverse reflections?  For example, you could include metrics quantifying the diversity of generated reflections compared to baseline methods. You can also explain the reason behind it.
To me, it is more like a magic prompt. Can you explain why this simple prompt works?
2. The whole framework is built on reflection, with Msr replaced by Mdr. Given my first comment and considering that utilizing memory to improve response diversity is widely explored, the contribution of DoT is limited.

**Questions:**

1. Line 700 has un-referred figure

---

> ### Author Response · Authors · 2024-11-13
>
> We thank the reviewer for their valuable feedback.
>
>
> **W1:** We clarify that our method does not claim to *guarantee diverse reflections* but instead aims to promote them (L::71-73). To quantify the diversity, we calculate the average pairwise cosine similarity across generated self-reflections for each problem within a dataset, reporting both the mean and standard deviation. We use the ‘all-Mini-LM-v6’ model from SentenceTransformers [1] to generate embeddings for these reflections. This metric is presented for HumanEval and LeetCodeHardGym across two different LLMs. A lower similarity score indicates greater diversity.
>
> To clarify further, in addition to using an explicit prompt, we also incorporate one-shot sampling to enhance the diversity of the generated self-reflections. Moreover, the prompt is supplemented by feedback from the environment, as well as context from previous implementations and self-reflections. A recent concurrent study [2] supports our approach, demonstrating that one-shot sampling can produce semantically diverse outputs. In the final version, we will include this diversity metric across all relevant tables and provide a citation for the referenced work.
> The tables below demonstrate that both DoT and DoT-Bank exhibit lower similarity scores compared to baseline methods, further supporting the effectiveness of our approach in reducing redundancy within reflections. Figure-4 corroborates this observation.
>
> -----
>
> |               | **HumanEval**        |        |
> | ------------- | ---------------- | ------ |
> | **Llama-3.1-70B** | **Similarity Score** | **Pass@1** |
> | LATS          | 0.65 (0.41)      | 84.76  |
> | Reflexion     | 0.83 (0.09)      | 87.2   |
> | DoT           | **0.48 (0.11)**      | **89.63**  |
> | DoT-Bank      | **0.48 (0.12)**      | **93.29**  |
> |               |                  |        |
> |               |                  |        |
> |               | **LeetCodeHardGym**  |        |
> | **Sonnet3.5**  | **Similarity Score** | **Pass@1** |
> | LATS          | 0.70 (0.29)      | 42.5   |
> | Reflexion     | 0.83 (0.13)      | 42.5   |
> | DoT           | **0.62 (0.13)**      | **50**     |
> | DoT-Bank      | **0.61 (0.10)**      | **52.5**   |
>
> -----
>
> **W2 (Concerns on Contributions):** We believe that we are the first to identify repetition in reflections as a limitation in self-reflection-based frameworks—a key contribution of our work. In Section 2.1, we clarify that DoT builds on Reflexion, but we observe that substituting M_sr with M_dsr to enhance diversity yields consistent improvements over both Reflexion and LATS across various base LLMs. Additionally, as shown in Algorithm 1 (lines 20-22), DoT differs subtly but critically: rather than generating a single reflection per iteration like Reflexion, DoT produces ‘k’ diverse reflections and selects the most promising one, effectively emulating a greedy BFS approach.
>
> Our Task Agnostic Memory Bank further complements DoT, enhancing generation quality. Section 4 provides a comparison with recent memory-based methods; if possible, we would appreciate specific references to any memory-related works the reviewer has in mind, as this could help further contextualize our contributions. We have also included code and logs of generated implementations and reflections in the supplementary files to allow for a qualitative examination of DoT's reflections.
>
> DoT variants address the identified shortcomings and consistently improve performance across various programming benchmarks. Notably, on the challenging LeetCodeHardGym dataset (introduced in Reflexion), where previous methods show minimal improvements over base LLMs, DoT variants achieve a significant performance boost across LLMs (GPT-o1, Sonnet-3.5, GPT-4o), with gains of up to 12.5%.
>
> **Q1.** Thank you for pointing this out. The missing reference in line 700 corresponds to Figure 7.
>
> -----
> References
>
> [1] https://sbert.net/docs/sentence_transformer/pretrained_models.html
>
> [2] How Far Can We Extract Diverse Perspectives from Large Language Models? Hayati. et. al.,  EMNLP 2024

---

> > ### Comment · Reviewer_7Wju · 2024-11-29
> > **Follow-up question**
> >
> > I sincerely thank the authors for the additional experiments on diversity to show that the magic prompt really works on generating diverse output. The experiment results seem promising, which addresses most of my concerns. However, can you elucidate why such simple prompting works?
> >
> > Sorry for the late reply.

---

> > > ### Author Response · Authors · 2024-11-29
> > >
> > > We sincerely thank the reviewer for their feedback and acknowledgment of the additional experiments demonstrating the effectiveness of our approach in generating diverse outputs. We would like to clarify a potential misunderstanding regarding the role of the prompting strategy in our method.
> > >
> > > **Diversity in DoT’s Reflections** The diversity in DoT’s generated self-reflections primarily arises from our one-shot sampling strategy, with the explicit prompting playing only a supporting role. The prompt used for the M_dsr agent is generic and not the result of prompt engineering or iterative refinement. Thus, our method does not employ any "magic prompt" to encourage diversity in the generated reflections.
> > >
> > > Unlike previous methods such as Reflexion and LATS, which generate reflections independently in separate LLM calls, DoT employs one-shot sampling (described in Lines 200–207 of our paper) to produce k-diverse reflections within a single LLM call. This approach inherently supports more semantically diverse outputs by providing just enough context to structure the responses while avoiding overfitting to repetitive patterns. Our empirical results, as well as the diversity metric ablation study, corroborate this hypothesis. Furthermore, recent concurrent work [1] aligns with our findings, suggesting that one-shot sampling can enhance the semantic diversity of outputs.
> > >
> > > **How DoT Differs from [1]** While [1] focuses on increasing output diversity through socio-demographic and criteria-based prompting or RLHF to align with human perspectives, our work is fundamentally distinct. DoT leverages diverse self-reflections not for ethical or social alignment but to enhance reasoning abilities. The iterative, diversity-driven refinement central to our framework focuses on improving logical reasoning, distinguishing our work from the broader considerations addressed in [1].
> > >
> > >
> > > We hope this response resolves any remaining concerns and are happy to provide further clarification or address follow-up questions.
> > >
> > > ----
> > >
> > > References
> > >
> > > [1] How Far Can We Extract Diverse Perspectives from Large Language Models? Hayati. et. al., EMNLP 2024

---

> ### Author Response · Authors · 2024-11-19
> **Following Up with the Reviewer**
>
> Thank you again for your review. As the discussion period is concluding soon, we would like to follow up to ensure we have sufficiently addressed your concerns and to kindly ask if you might consider reassessing your evaluation.
>
> To summarize our updates:
>
> **1. Updated Section 3.3.1.** We incorporated a diversity metric and referenced recent related work that supports our observation that one-shot sampling can produce semantically distinct outputs.
>
> **2. Contributions and Analysis.** One of the key contributions of our work is the well motivated analysis to identify repeated reflections as shortcoming and using diverse reflections to alleviate it. We included a new study in Appendix A.2.5 to show that naively increasing total generated reflections in Reflexion has minimal impact on performance. Appendix A.2.1 shows that our reported gains are statistically significant. We hope that these new experiments and clarifications make our contributions clear.
>
> We hope these additions clarify our contributions and address your concerns. We would be happy to answer any follow-up questions and address any remaining concerns.

---

> > ### Author Response · Authors · 2024-11-24
> > **Final Follow-Up on Reviewer Feedback**
> >
> > As the rebuttal phase nears its conclusion, we wanted to follow up to ensure our responses and updates have addressed your concerns and to kindly request that you consider reassessing your evaluation. We would be happy to answer any follow-up questions and address any remaining concerns.

---

> > > ### Author Response · Authors · 2024-11-27
> > >
> > > Dear Reviewer 7Wju,
> > >
> > > We kindly request your feedback on our rebuttal and updated PDF. We believe our responses and revisions have thoroughly addressed the concerns you initially raised, and we are eager to clarify or resolve any remaining concerns.
> > >
> > > **We would greatly appreciate it if you could kindly reassess your evaluation.**
> > >
> > > Thank you for your time and consideration.

---

> ### Author Response · Authors · 2024-12-02
> **Follow-up**
>
> Dear Reviewer 7Wju,
>
> We hope our response has adequately addressed your concerns. As the discussion period nears its conclusion, we kindly ask if you have any remaining questions or if you could reassess our paper based on the clarifications provided.
>
> Thank you for your time and consideration.

---

### Author Response · Authors · 2024-11-18
**Revised Paper Summary**

We sincerely thank the reviewers for their insightful feedback, which has significantly improved our work. We have addressed all concerns through additional experiments, clarifications, and improvements to the paper. Below is a summary of the key updates, highlighted in blue in the revised version:

1. **Introduction**: Added a connecting paragraph in Section I, citing a relevant recently accepted work to strengthen support for our memory-bank approach (insights from previous tasks are helpful) and improve the narrative coherence.
2. **Quantitative Analysis**: Introduced a diversity metric for generated reflections in Section 3.3.1 and cited recent work validating the efficacy of one-shot sampling for diverse outputs.
3. **Implementation Details**: Expanded details about the memory-bank implementation (Appendix A.1) and explicitly documented hyper-parameters in Section 3.1.
4. **New Experiments**: Conducted a study showing minimal performance gains from naively increasing the number of reflections (Appendix A.2.5) and repeated a subset of our experiments three times to ensure statistical significance, with results detailed in Appendix A.2.1. Token usage analysis is provided in Appendix A.2.4.
5. **Citations**: Included references to missing related works, such as Self-Refine and other recent publications.

We believe these updates comprehensively address the reviewers' concerns and improve the quality and clarity of the paper. We welcome any further questions and kindly request your favorable consideration.

---

### Author Response · Authors · 2024-12-03
**Rebuttal Summary**

We thank the reviewers for their feedback and valuable discussions.

We are encouraged by their recognition of the clear and concrete motivation for our work (**7Wju, vf4f, vtDD, RxnW**), the modular design of our framework (**vf4f**), the sound experimental methodology with consistent performance improvements (**vf4f, vtDD**), and the practical, generalizable nature of our contributions (**vf4f**).
While Reviewers **7Wju** and **RxnW** noted that diversity has been widely explored, *we clarified that our work uniquely focuses on diversity in reflections to enhance reasoning in iterative frameworks.* Unlike prior work, which emphasizes diversity in generic response generation, our novel combination of a diverse-reflections module and a task-agnostic memory bank enables deeper decision-space exploration and better task generalization. This positions our work as a meaningful contribution that extends existing reasoning frameworks in a previously unexplored direction.
Below, we outline how we addressed key concerns raised during the rebuttal:
1. **Diversity in Reflections** (**7Wju, vf4f**):
We introduced a diversity metric based on average pairwise cosine similarity to empirically validate the diversity of DoT reflections (```Section 3.3.1```). Results show DoT achieves significantly lower similarity scores compared to baselines like LATS and Reflexion, demonstrating improved exploration capabilities.
2. **Task-Agnostic Memory Bank Justification** (**RxnW**):
We extended experiments to validate the memory bank’s utility in enhancing cross-task generalization. By integrating relevant trajectories based on cosine similarity, the memory bank boosts performance. Ablation studies (```Table 9```) comparing random versus contextually similar trajectories further highlight its effectiveness.
3. **Statistical Significance** (**vtDD**):
We repeated key experiments to ensure statistical significance (```Appendix A.2.1```). This reinforces the robustness of our results and addresses concerns about random variations.
4. **Efficiency Analysis** (**vtDD, RxnW**):
We clarified token usage (```Appendix A.2.4```) and showed that one-shot sampling achieves superior results cost-effectively compared to iterative sampling. Detailed cost analyses, including DoT-Bank, demonstrate the resource efficiency of our approach.
5. **Technical Details and Clarity** (**RxnW, vtDD, vf4f**):
We expanded on memory bank construction (```Appendix A.1```), clarified hyperparameters (```Section 3.1```), and refined the narrative with additional citations. Results have been reformatted for clarity, and citation issues have been resolved.

We believe our extensive experiments and detailed responses have addressed their concerns, and we kindly request their favorable consideration of our submission.

---

### Meta-Review · Area_Chair_FW1k · 2024-12-23

**Metareview:**

Given the well-motivated problem and the practical, generalizable nature of the contributions, despite the limited novelty, I recommend accepting this paper with the understanding that it is a borderline case. The authors have addressed the key concerns raised during the rebuttal process, and their extensive experiments and detailed responses have strengthened the paper's case for acceptance. However, it is advised that the authors continue to work on differentiating their contribution from existing literature and provide a more robust theoretical foundation for their approach in future work.

**Additional Comments On Reviewer Discussion:**

The authors expressed gratitude for the reviewers' feedback and discussed how they addressed the key concerns raised:

Diversity in Reflections: To validate the diversity of DoT reflections, a diversity metric based on average pairwise cosine similarity was introduced, showing significant improvement over baselines like LATS and Reflexion.

Task-Agnostic Memory Bank Justification: Additional experiments were conducted to demonstrate the memory bank's role in enhancing cross-task generalization, with ablation studies highlighting its effectiveness.

Statistical Significance: Key experiments were repeated to ensure statistical significance, reinforcing the robustness of the results.

Efficiency Analysis: Clarifications on token usage were provided, and it was shown that one-shot sampling is more cost-effective than iterative sampling.

Technical Details and Clarity: The authors expanded on memory bank construction, clarified hyperparameters, and refined the narrative with additional citations to improve clarity.

Key Points Raised by Reviewers and Authors' Responses:

Reviewer 7Wju noted the need for empirical evidence of diverse reflections. The authors responded by introducing a diversity metric and additional experiments, elucidating that diversity arises from one-shot sampling rather than a "magic prompt."

Reviewer RxnW questioned the memory bank's justification and the comparison with Reflexion. The authors added a connecting paragraph in the introduction, included a study showing minimal performance gains from increasing reflections, and provided a diversity metric to address these concerns.

Reviewer vtDD raised concerns about the practical usability and core contribution of diverse thought aiding LLM reasoning. The authors acknowledged the concerns and provided comparisons with current SOTAs, addressing the need for more comprehensive analysis.

Reviewer vf4f pointed out minor weaknesses, such as clarity in trajectory retrieval and t-SNE evaluation. The authors' revisions aimed to clarify these points and improve the overall presentation.

Despite the well-motivated problem and the practical, generalizable nature of the contributions, the limited novelty of the approach remains a key concern. Nevertheless, I recommend accepting this paper, recognizing that it is a borderline case. The authors have made valuable revisions in response to the key issues raised during the rebuttal process. Their extensive experiments and thoughtful responses have certainly strengthened the paper. However, to ensure greater distinction from existing literature and enhance the theoretical foundation of their approach, further work will be necessary in future iterations. It is suggested that the authors continue to refine and differentiate their contribution as they advance this research.

---

### Decision · Program_Chairs · 2025-01-22

Accept (Poster)